

# SMOS L-VOD shows that post-fire recovery of dense forests is slower than what is depicted with X- and C-VOD and optical indices

Emma Bousquet[1], Arnaud Mialon[1], Nemesio Rodriguez-Fernandez[1], Stéphane Mermoz[2], and Yann Kerr[1]

[1]Centre d'Etudes Spatiales de la Biosphère (CESBIO), Université de Toulouse (CNES/CNRS/INRAE/IRD/UPS), 18 av. Edouard Belin, bpi 2801, 31401 Toulouse CEDEX 9, France
[2]GlobEO, 31400 Toulouse, France

*Correspondence to*: Emma Bousquet (emma.bousquet@cesbio.cnes.fr)

**Abstract.** Anthropogenic climate change is now considered to be one of the main factors causing an increase in both frequency and severity of wildfires. These fires are prone to release substantial quantities of $CO_2$ in the atmosphere and to destroy natural ecosystems while reducing biodiversity. Depending on the ecosystem and climate regime, fires have distinct triggering factors and impacts. To better analyse and describe fire impact on different biomes, we investigated pre and post fire vegetation anomalies at global scale. The study was performed using several remotely sensed quantities ranging from optical vegetation indices (the enhanced vegetation index (EVI)) to vegetation opacities obtained at several microwave wavelengths (X-band, C-band, and L-band vegetation optical depth (X-VOD, C-VOD, and L-VOD)), ranging from 2 to 20 cm. It was found that C- and X-VOD are mostly sensitive to fire over low vegetation areas (grass and small bushes) or over tree leaves; while L-VOD depicts better the fire impact on tree trunks and branches. As a consequence, L-VOD is probably a better way of assessing fire impact on biomass. The study shows that L-VOD can be used to monitor fire affected areas as well as post-fire recovery, especially over densely vegetated areas.

## 1 Introduction

Wildfires are known to have several negative effects on soil and vegetation properties. They cause deterioration of soil structure and porosity, removal of organic matter, loss of nutrients, ash entrapment, decreasing of microbial and soil-dwelling invertebrate communities, etc. (Certini, 2005). By removing plant cover and inducing soil water repellency, wildfires can cause excess runoff which, in turn, can lead to floods and erosion (Moody and Martin, 2001; Shakesby and Doerr, 2005). Post-fire vegetation regeneration highly depends on the ecosystem and on the fire severity (Chu and Guo, 2013). In the Central Brazilian Amazon



forest (Amazônia legal), based on field measurements, Barlow et al. (2003) reported that live above-ground biomass (AGB) decreased by ~ 90 Mg ha$^{-1}$ one year post-burn, with a majority of small stems affected, and by ~ 195 Mg ha$^{-1}$ three years post-burn, with large trees suffering the highest mortality. Silva et al. (2018) found that wildfires can significantly reduce AGB of humid tropical forests for decades by amplifying tree mortality. Authors found AGB levels of burned field data ~ 25% below those of unburned control plots after 31 years.

Using remote sensing data and a forest growth model, de Faria et al. (2021) found a 19% decrease of the Amazonian forest AGB after fire, and a 44 year recovery time. However, despite common perceptions, some vegetation communities have adapted to such extreme conditions. Some coniferous trees (e.g. jack pine, black spruce) evolved to become fire resistant and to use the flames as a means for spreading their seeds, as the heat causes the opening of cones (Weber and Stocks, 1998). In South-East Australia, eucalyptus

dominated resprouter communities showed a rapid post-fire response and recovery rate in Sydney's drinking water supply catchments (Heath et al., 2016). These species are able to survive fire by activating dormant vegetative buds to produce regrowth. In savannas, recurrent fires help maintaining the structure, species composition, and biological diversity (Menaut et al., 1990). They prevent savannas from evolving towards woodlands (Gillon, 1983). In forests, prescribed burning is used to reduce hazardous accumulations of fuel,

and thus the risk of damage from wildfires (Sackett, 1975). These fires can even be necessary for canopy regeneration. For instance, a decline of sequoias population was observed when fires were suppressed in California (Parsons and DeBenedetti, 1979). Vegetation can thus recover from fire and if plants succeed in promptly recolonising the burnt area, the pre-fire level of most properties can be recovered and even enhanced (Certini, 2005).

Fires are a natural part of many ecosystems, being historically triggered by lightning strikes (de Groot et al., 2013). Nevertheless, in recent years, and in spite of various efforts, wildfires were proven to increase both in frequency and in severity worldwide. Summer 2021 saw an unprecedented number of fires around the Mediterranean Sea, in Siberia and in North America (CAMS, 2021). They often went out of control, took lives, destroyed properties and ecosystems, endangered species, and released unprecedented quantities of $CO_2$ in

the atmosphere (C3S, 2021; CAMS, 2021). The 2020 fire season was a record-setting year of wildfires in California and more generally in the Western US, linked with historically unprecedented level of vegetation dryness (Higuera and Abatzoglou, 2020). In southern Australia, the 2019–2020 extremely severe wildfires also became historically significant (Ehsani et al., 2020). In tropical rainforests, the Amazon in particular, wildfires have become increasingly prevalent over the past decades due to more frequent droughts and periodic El

Niño events (Aragão et al., 2018; Chen et al., 2013; Cochrane, 2003; Jolly et al., 2015), but also to forest disturbances such as selective logging and deforestation, that lead to forest desiccation and reduce rainfall (Aragão, 2012; Asner et al., 2010). All these factors may strongly influence the future stability of tropical



forests (de Faria et al., 2017). The Amazon could move from a net carbon sink to a net carbon source (Brando et al., 2020). These fire-prone conditions are largely attributable to anthropogenic climate change, and to human pressure (Weber and Stocks 1998; Randerson et al., 2006; Jin et al., 2012; Masrur et al., 2018; Goldammer and Crutzen, 1993).

Naturally, understanding the potential factors triggering large uncontrollable fires is important to anticipate them and thus to reduce their impacts. Mhawej et al. (2015) presented and categorized 28 wildfire likelihood factors into climatic (e.g. precipitation, temperature, air humidity, wind speed), topographic (e.g. slope, altitude), in situ (e.g. fuel type, soil texture, tree diameter), historical, and anthropogenic factors. Drought, i.e. the concomitant increase of air dryness and decrease of fuel moisture, was identified as the most significant fire likelihood factor (Ray et al., 2005). Indirectly, drought also reduces soil moisture, leading to leaf shedding and branch losses (Pausas and Bradstock, 2007). This process increases fuel accumulation and direct sunlight reaching the forest floor, increasing forest flammability (Nepstad et al., 2001). Surveying the soil moisture and the biomass status could then be a good indicator for fire detection. Currently, drought is mainly monitored with temperature and precipitation observations. Fuel availability is monitored with optical sensors, but with fast saturation over dense vegetation, and lack of sensitivity for dry biomass (not green), which is the main fuel.

Chu and Guo (2013) have reviewed the use of remote sensing in addition to field campaigns to evaluate the impacts of fire and post-fire vegetation regeneration. Indicators and metrics based on multispectral satellite imagery (visible and infrared) are the most frequently used, such as the normalized difference vegetation index (NDVI), the enhanced vegetation index (EVI), and the normalized burned ratio (NBR) (Pérez-Cabello et al., 2021). Despite their fast saturation over dense forests, they still provide a good proxy for green vegetation regrowth. Microwave data also shown a good potential to monitor post-fire biomass recovery. Mermoz and Le Toan (2016) used L-band ALOS PALSAR to assess forest regrowth in South-East Asia. Fernandez-Carrillo et al. (2019) used active L-band SAR data to estimate the tree survival in eucalyptus forests of Western Australia. Zhang et al. (2021) analysed forest canopy dynamics in the southern Amazon during the 2019 fire season using passive microwave vegetation optical depth at C-band (C-VOD). Authors found a larger magnitude and a faster recovery of optical-based indices (NDVI, EVI, NBR) than C-VOD.

With the arrival of L-band radiometers such as the Soil Moisture and Ocean Salinity (SMOS) satellite, it is possible to infer complementary information, typically surface soil moisture, biomass (i.e. fuel) and its water content. The rationale of this study is to investigate how L-band radiometry responds to fire events over different ecosystems and under several climate circumstances. SMOS satellite has been operating for almost 12 years now and we have access to a large catalogue of major fires. This study also presents for the first time L-band used in conjunction with other sensors, from optical (EVI) to X- and C-bands. All the vegetation





variables (VVs) derived from various sensors then represent different aspects of vegetation structure: canopy greenness for EVI, top canopy biomass for X-VOD (leaves and small branches), deeper canopy biomass for C-VOD (larger branches), and the whole AGB layer for L-VOD (stems and trunks) (Guglielmetti et al., 2007; Frappart et al., 2020). The complementarity of the derived VVs along with climate variables (CVs) (air temperature (T), precipitation (P), soil moisture (SM), and terrestrial water storage (TWS)) was used to identify the fire likelihood factors, and the immediate and long-term fire impacts on vegetation. To do this, we first analysed three particular cases of large fires in various environments and then extended the analysis to the global scale.

## 2 Data

### 2.1 Fires

Fires were obtained from the National Aeronautics and Space Administration (NASA) MODerate resolution Imaging Spectroradiometer (MODIS) Active Fire product (MOD14A1_M). The product is a quantification of the number of fires observed within a 1000 km² area over a month. A fire must cover at least ~ 1000 m² to be detected. The Active Fire product is based on the 1 km fire channels at 3.9 and 11 μm of MODIS Terra and Aqua satellites (Justice et al., 2006). It is distributed at 0.1 deg resolution and at a monthly time scale by NASA Earth Observations (NEO) portal.

### 2.2 Precipitation

Precipitation (P) data come from the Precipitation Estimation from Remotely Sensed Information using Artificial Neural Networks- Climate Data Record (PERSIANN-CDR). The precipitation estimate uses the PERSIANN algorithm on GridSat-B1 infrared satellite data, and training of the artificial neural network on the National Centers for Environmental Prediction (NCEP) hourly precipitation data (Ashouri et al., 2015). The dataset is distributed by National Oceanic and Atmospheric Administration (NOAA) at a daily time scale, and at 0.25 deg resolution in the latitude band 60°S – 60°N.

### 2.3 Soil Moisture

The soil moisture (SM) dataset comes from SMOS satellite, launched by the European Space Agency (ESA) in 2009 (Kerr et al., 2001). It performs passive measurements of the thermal emission of the Earth at L-band (1.4 GHz, 21 cm). L-band VOD and SM are derived from SMOS brightness temperatures using the L-band Microwave Emission of the Biosphere (L-MEB) radiative transfer model (Wigneron et al., 2007; Kerr et al., 2012). L-band SM is the volume of water per volume of soil ($m^3 m^{-3}$) in the top surface soil layer (~ 5 cm). The





footprint size is ~ 43 km in average (Kerr et al., 2010). We considered the ESA level 2 SM dataset in version 7.2 (L2 v720) resampled to the global cylindrical Equal-Area Scalable Earth (EASE) Grid version 2.0 (Brodzik et al., 2012) at 625 km² spatial sampling (25 km × 25 km at 30 deg of latitude). Ascending (6 am) and descending (6 pm) overpasses were averaged, from June 2010 to December 2020.

## 2.4 Terrestrial Water Storage

Terrestrial water storage (TWS) anomalies from the Gravity Recovery and Climate Experiment (GRACE) satellite were also considered. We used monthly GRACE/GRACE-Follow On (FO) Level-3 product provided through the Gravity Information Service (GravIS) web portal of the German Research Centre for Geosciences (GFZ) at 1 deg latitude-longitude grids (Boergens et al., 2019). TWS anomalies represent the water mass anomalies from snow, surface water, soil moisture, and deep groundwater. They are derived from

measurements of temporal changes in the Earth's gravity field. Data were lacking for 35 dates of the ten-year dataset. One-time gaps were filled by linear interpolation; consecutive missing months were not considered.

## 2.5 Temperature

Temperature (T) data come from the Land Surface Temperature (LST) dataset from MODIS Terra satellite (NASA). Daytime and night time measurements were averaged (MOD11C3 Version 6 product in a Climate

Modeling Grid (CMG), LST_Day_CMG and LST_Night_CMG, Wan et al., 2015). These datasets are obtained using MODIS thermal infrared bands from 3 to 15 μm, and distributed by NASA Land Processes Distributed Active Archive Center (LP DAAC) at a monthly time scale and at 0.05 deg resolution.

## 2.6 Vegetation Optical Depth

Vegetation optical depth (VOD) is a remotely sensed indicator related to AGB and to vegetation water content

(VWC) (Kerr and Njoku, 1990; Jackson and Schmugge, 1991; Jones et al., 2011; Rahmoune et al., 2014; Vittucci et al., 2016; Rodriguez-Fernandez et al., 2018; Mialon et al., 2020; Konings et al., 2021). The lower frequencies have better capabilities to penetrate deeper within the canopy (Ulaby et al., 1981). At L-band, VOD is sensitive to coarse woody elements, such as trunks, stems, and branches. At C- and X-band, VOD is more sensitive to thin stems and leaves (Guglielmetti et al., 2007). In this paper, L-VOD comes from SMOS level 2 dataset in

version 7.2 (L2 v720) measured at 1.4 GHz (λ = 21 cm), resampled to EASE-Grid 2.0 at 625 km² resolution (25 km × 25 km at 30 deg of latitude). In the SMOS retrieval algorithm, the vegetation attenuation is taken into account by the τ parameter of the τ − ω model (Mo et al., 1982) which corresponds to the L-VOD. Data from June 2010 to December 2020 were considered, and ascending (6 am) and descending (6 pm) overpasses were averaged. C- and X-VOD from the Japan Aerospace Exploration Agency (JAXA) Global Change Observation





Mission (GCOM) Advanced Microwave Scanning Radiometer (AMSR)-2 dataset were also considered (Imaoka et al., 2010). C- and X-VOD are measured at 6.9 GHz (λ = 4.3 cm) and 10.7 GHz (λ = 2.8 cm) respectively. C2-band (7.3 GHz, λ = 4.1 cm) was not discussed in this paper as the data were mostly redundant with C1-band (6.9 GHz). We used the daily L3 V001 VOD products, from July 2012 to December 2020, processed with the Land Parameter Retrieval Model (LPRM) algorithm (Owe et al., 2008) and distributed by NASA on a regular grid

at 25 km × 25 km resolution. Ascending (1:30 pm) and descending (1:30 am) overpasses (LPRM_AMSR2_A_SOILM3 and LPRM_AMSR2_D_SOILM3) were averaged.

## 2.7 Enhanced Vegetation Index

VODs values were compared with the optical-based enhanced vegetation index (EVI) from MODIS (NASA) MOD13C2 and MYD13C2 Version 6 for Aqua and Terra Satellites respectively, distributed at 5600 m resolution

(Didan, 2015). EVI represents canopy greenness, with an improved sensitivity over high AGB regions compared to NDVI. It is obtained by combining measurements at red (λ = 0.6−0.7 μm, f ~ 460 THz) and near infrared wavelengths (λ = 0.7−1.1 μm, f ~ 330 THz).

## 2.8 Auxiliary data

### 2.8.1 Year of gross forest cover loss event

The year of gross forest cover loss event map (the so-called lossyear product) from Hansen et al. (2013) was used to observe the forest loss rate and year within a SMOS pixel, for the three major fires studied (Fig. 2). This map represents the first year of detected tree loss during the period 2000–2020, defined as a stand-replacement disturbance, or a change from a forest to non-forest state. This dataset is based on Landsat images and is distributed at ~ 30 m resolution with 10x10 square degree tiles at

https://glad.earthengine.app/view/global-forest-change. Each year of the period 2010−2020 was extracted from the forest loss product and averaged into SMOS EASE-Grid 2.0, so as to obtain a yearly percentage of forest loss.

### 2.8.2 Land cover

A land surface climatology map based on 10 years (2001–2010) of the MODIS MCD12Q1 product at 500 m

resolution (Broxton et al., 2014) was used to filter the data and to distinguish four different vegetation types (see Sect. 3). This land cover map allows to identify 17 ecosystems based on the IGBP (International Geosphere-Biosphere Programme) class labels.



### 2.8.3 Above-Ground Biomass

The global map of AGB (Mg ha$^{-1}$) from Santoro et al. (2021) was used to distinguish sparse from dense forests
(see Sect. 3.2). This map is distributed through the ESA Climate Change Initiative (CCI) Biomass at 100 m
resolution. It combines a large pool of spaceborne remote sensing observations from two synthetic aperture
radar (SAR) missions (Envisat and ALOS), and uses optical (Landsat) and LiDAR (Icesat GLAS) data to support
the model calibration procedure. The ESA CCI Biomass map representative of 2010 was used here because it
provides an AGB information prior to the studied fire events (2011–2020).

### 2.8.4 Snow and Ice

The Interactive Multisensor Snow and Ice Mapping System (IMS) database was used to mask areas covered
with snow or ice (see Sect. 3.2). We used the IMS Daily Northern Hemisphere Snow and Ice Analysis at 4 km
resolution, version 1 (Helfrich et al., 2007), provided by the National Snow and Ice Data Center (NSIDC).

### 2.8.5 Flooding

Flooded areas were filtered out (see Sect. 3.2) based on the Global Inundation Estimate from Multiple
Satellites (GIEMS-2) dataset (Prigent et al., 2019). It provides long-term monthly estimates of surface water
extent, including open watern wetlands, and rice paddies. The methodology combines passive and active
microwaves, visible and near-infrared observations (SSM/I, ERS, AVHRR). The water fraction is delivered
globally from 1992 to 2015, on an equal area grid of 0.25 deg × 0.25 deg at the equator (~ 28 km × 28 km).
Flooded areas were detected with a climatology over the 1992–2015 period.

### 3. Methods

First, we investigated three regional tests cases, diversely affected by human activities, from dry to wet
climate, which recently suffered from a severe fire. These areas consist in: i) a eucalyptus open forest in a
human-affected environment, under dry El Niño conditions in Australia; ii) a Mediterranean climate with
needleleaf forests, agriculture, and human activities in California; and iii) a very dense primary forest in a
tropical wet climate with no human activities in Amazonia (see Sect. 3.1). Following this first step, the study
was extended to the global scale, for five vegetation types, by selecting the major fires of the last decade. The
rationale was to capture significant events occurring over an area large enough to be observed with the SMOS
satellite without any ambiguity.

Four CVs related to the fire risk were considered: precipitation, SM, TWS, and temperature. Wind is another
predominant fire likelihood factor (Albini, 1993), but was not studied here due to the lack of global reliable





database at a monthly time-scale. Vegetation status before, during, and after fire was monitored with four optical and microwave variables: EVI, X-VOD, C-VOD, and L-VOD. Data from June 2010 to December 2020 were considered (10.5 years), except for C- and X-VOD from AMSR-2 which were only available from July 2012.

Monthly averages of all datasets were computed and resampled to SMOS grid. To do so, they were first oversampled to 1 km resolution and then averaged to SMOS EASE-Grid 2.0 (~ 25 km resolution) with GDAL (GDAL/OGR contributors, 2020).

### 3.1. Case study : analysis of three major fires

### 3.1.1. Wildfires in the South Coast of New South Wales in Australia

The first studied area is located in the South Coast of New South Wales in Australia, between [33.53°S – 37.72°S] and [149.40°E – 150.17°E] (Fig. 1) and covers thirteen SMOS pixels. The dominant vegetation type is eucalyptus open forest (McColl, 1969; DEWR 2007). The climate is warm temperate with dry summer (Kottek et al., 2006). The mean rainfall is ~ 1000 mm year$^{-1}$, and the mean temperature is ~ 15°C (McColl, 1969). The topography varies between 0 to 600 m above sea level. The 2019–2020 wildfires in Australia were influenced

by El Niño Southern Oscillation (Dowdy, 2018). They became historically significant as they were widespread and extremely severe, in particular in New South Wales (Ehsani et al., 2020). The tree cover loss map (Hansen et al., 2013) indicates a 25% forest loss in 2020 in the studied area (Fig. 2).

### 3.1.2. Mendocino Complex fire in California

The second studied area is located in California, between [38.96°N – 39.46°N] and [122.68°W – 123.20°W] (Fig.

1). It corresponds to four SMOS pixels. The area is covered with evergreen needleleaf forest and woody savannas (Broxton et al., 2014), and is much urbanised. The climate is warm temperate (Kottek et al., 2006), with dry, windy, and often hot weather conditions from spring through late autumn that can produce moderate to severe wildfires (Crockett and Westerling, 2018). The 2018 fire season was the most extreme on record in Northern California (now second to the 2020 fire season) in terms of number of fatalities, destroyed structures,

and burned areas (Brown et al., 2020). The largest fire complex in state history, the Mendocino complex, burned nearly 186000 ha of vegetation between July and September 2018. The Mendocino complex started on July 27 near Lakeport lake and included two wildfires: the Ranch fire at the North of the lake, which was the largest single fire in state history, and the River fire at the West of the lake. The River Fire was fully contained on August 13 after having burnt 198 km²; and the Ranch fire was contained on September 18 after burning

1660 km² (estimation from Bureau of Land Management). This wildfire was responsible for a 34% forest loss in this region (26% in 2018 and 8% in 2019, Fig. 2), and caused a burn severity ranging from moderate to high.



### 3.1.3. Santarem wildfire in the Amazon

The third studied area is located in the Amazon rainforest near Santarem city (Brazil), between [3.14°S – 2.75°S] and [53.95°W – 54.13°W] (Fig. 1) and covers two SMOS pixels. The evergreen broadleaf forest is dense

(L-VOD = 1.02; AGB = 280 Mg ha$^{-1}$ in average over the area). The climate is hot and humid, with annual mean temperature of 25°C and mean precipitation of 1920 mm year$^{-1}$ (Berenguer et al., 2018). During the strong El Niño event in December 2015, a severe drought caused large fires in this area, with no link with anthropic deforestation (Berenguer et al., 2018). They induced a 20% forest loss in 2016 in the studied area (Fig. 2).

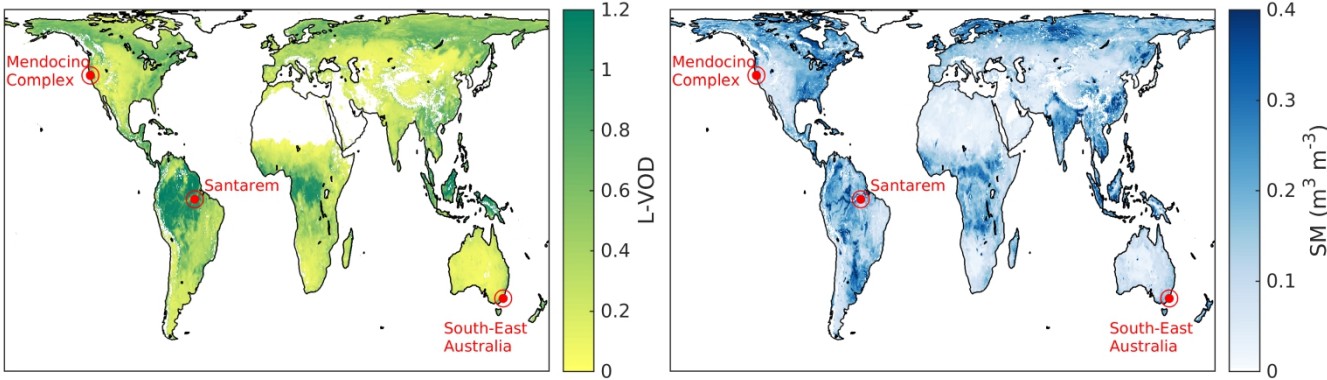

**Figure 1 - Global maps of SMOS L-VOD (left) and SM (right), in average for 2011–2020. The red dots show the locations of the three areas of interest: the Mendocino complex in California, Santarem in Amazonia, and the South Coast of New South Wales in Australia.**

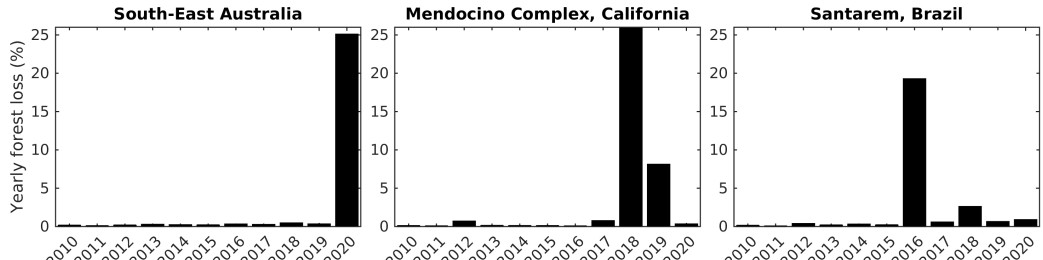

**Figure 2 – Yearly forest loss (%) attributed to the three burnt areas under study, from Hansen et**
**al. "lossyear" product.**

EVI, X-, C-, L-VOD, P, SM, TWS, and T anomalies time series were plotted over the three studied sites. The anomaly time series of a variable x is the difference between the original time series and the mean climatology, which is the monthly long-term mean of this variable. They are defined as:

$$anom(x(t)) = x(t) - climatology(x(m)) \qquad (1)$$

and

$$climatology(x(m)) = \frac{1}{y_n}\sum_{y=1}^{y_n} x(m + (y-1)*12) \ , \ \forall m = 1:12 \qquad (2)$$





where t is the month number from January 2010 (6 to 132 in this study); m is the month of the year, between 1 and 12, with m = t mod 12 if t mod 12 > 0, and m = 12 if t mod 12 = 0; y is the year number, from 1 to $y_n$, with $y_n$ = 11 here as the climatology was computed on the period 2010–2020. Plotting the anomalies time
series enables to remove the natural seasonal cycle so as to observe only the variations due to specific events. The average pre-fire variable value was subtracted from the anomalies time series, only if at least twelve months of data were available before the fire event. It enables to observe the anomalies with respect to the pre-disturbance value.

## 3.2. Extension to the global scale

Fires were then studied at the global scale to assess the general factors related to fire risk and the response of vegetation to fire according to the ecosystem. Several areas were excluded: i) snow-covered months (20% maximum monthly snow coverage), because snow interferes with most satellite observations; ii) flooded areas, with a maximum threshold of 10% on GIEMS-2 water fraction climatology, because standing water perturbs VOD measurements (Bousquet et al., 2021); iii) Australia because numerous and severe events took place in
2019/2020, which prevailed over the global dataset and prevented to perform a robust post-fire study. The global scale was divided into five land cover classes: needleleaf forests (IGBP labels 1 and 3), sparse broadleaf forests (IGBP labels 2 and 4, AGB ≤ 150 Mg ha$^{-1}$), dense broadleaf forests (IGBP labels 2 and 4, AGB > 150 Mg ha$^{-1}$), tropical and subtropical savannas (IGBP labels 8 and 9, latitudes between 40°S and 40°N), and grasslands (IGBP label 10). Only the range July 2012–December 2020 was conserved here for all datasets so
as to match with AMSR-2 time-period. Fires with a number larger than 5 in the MODIS dataset were considered, and only if no other fire occurred on the same area (number lower than 2 apart from the main fire event). This was done to observe only the impact of the major fires, without any other disturbance. These thresholds were tested and determined empirically. CVs and VVs anomalies were computed with Eq. (1) and (2), with a climatology over all dates excepted the year of the fire (from -5 to +6 months from the fire date $t_{fire}$), in order
to remove these exceptional values. The anomalies time series were then shifted to collocate in time all fire events. Averaged anomalies for a month m were kept only if at least one quarter of the dataset was available for that month m, in order to be representative of the global dataset.

CVs pre-fire anomalies and VVs post-fire anomalies were also aggregated at different time frames and plotted by variable, in order to compare their pre and post fire temporal behaviour in different ecosystems.
The standard error of the mean of the measurements σ was also computed with:

$$\sigma(p) = \frac{std(p)}{\sqrt{n}}$$
(3)

where std is the standard deviation of the population p and n is the number of samples.



# 4. Results

## 295   4.1. Case study: analysis of three major fires

**Figure 3 – EVI, X-, C-, L-VOD, P, SM, TWS, and T anomalies time series on (a) South-East Australia (13 SMOS pixels), (b) the Mendocino Complex, California (4 SMOS pixels), and (c) Santarem (2 SMOS pixels).**





In evergreen forests of the South Coast of the New South Wales region in Australia (Fig. 3a), fires are associated with high temperature and low precipitation (anom(T) = +3°C, anom(P) = -80 mm). The drought started 3 years before fire (decrease in precipitation, SM, and TWS). All VVs exhibit the same pattern, which is i) a constant and mild decrease since 2012, linked with the afore mentioned drought; ii) a strong decrease during the fire event (anom(VV) ~ -0.15); and iii) a rapid post-fire recovery (~ 1 year). C-VOD is the most
affected VV.

   In California, no major pre-fire drought is visible in summer 2018 (Fig. 3b). The Mendocino Complex provoked a decrease in all VVs, particularly in L-VOD (anom(L-VOD) = -0.08) and in EVI (anom(EVI) = -0.10). C- and X-VOD regained rapidly (~ 1 year) their pre-fire values, but EVI and L-VOD did not.

   The dense and humid rainforest near Santarem (Brazilian Amazon) shows stable VVs values before the
December 2015 fire (Fig. 3c). L-VOD signal is quite noisy because only two SMOS pixels were considered here. Strong positive temperature anomalies (+3°C), negative precipitation anomalies (-160 mm) and TWS anomalies (-60) are visible before and during the fire. Surprisingly, SM stayed stable during the fire. L-VOD was substantially impacted by the fire (anom(L-VOD) = -0.14), as well as EVI (anom(EVI) = -0.09). C- and X-VOD were barely affected (anom(C-VOD) = -0.04, anom(X-VOD) = -0.01). EVI recovered in ~ 2–3 years, whereas L-
VOD never recovered its pre-fire level.

## 4.2. Extension to the global scale

In this section, the major fires from July 2012 to December 2020 were analysed at the global scale, by shifting the anomalies time series of all variables on the fire date $t_{fire}$. The considered fires are well spread spatially and temporally over the nine-year period (Fig. 4). In savannas and grasslands (Fig. 5a and 5b), pre-fire
hydrologic variables are stable (positive for grasslands) and temperature anomalies are negative during 2 years before fire. Concurrently, VVs start to increase and reach a maximum a few months before the fire event (particularly C- and X-VOD over grasslands). The vegetation material surplus implies an increase of the available fuel, which may facilitate the propagation of wildfires. VVs anomalies also show a light surplus over needleleaf forests just before the fire event (Fig. 5c). Over broadleaf forests (Fig. 5d and 5e), a pre-fire drought
is visible through the temperature increase and the precipitation, SM, and TWS decrease, 12 months pre-fire for sparse forests, 8 months pre-fire for dense forests. For all ecosystems, these drought conditions intensify just before and during fire, and end a few months after fire. During fire, all VVs abruptly decrease in all ecosystems, EVI being the most impacted one, excepted over dense forests where L-VOD heavily decreases (Fig. 5e). For all ecosystems, EVI recovers more rapidly than VODs. L-VOD is particularly long to recover over
dense broadleaf forests (more than 4 years, Fig. 5e). Needleleaf forests (Fig. 5c) exhibit a slow recovery time for all VVs, with ~ 3 years for EVI and ~ 4 years for VODs. VODs even continue to decrease during 1 year post-

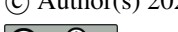

fire in this ecosystem. In low vegetation ecosystems (Fig. 5a and 5b), C- and X-VOD never regain their immediately pre-fire values, which were particularly high.

CVs anomalies were also averaged in space and in time, within time frames of 6 months, from 24 to 1 month pre-fire, in order to observe their general trends (Fig. 6). The error bars were computed with Eq. (3). Precipitation anomalies (Fig. 6a) are negative from 6 months pre-fire for all classes, and reach -10 mm month$^{-1}$ in average before the fire event. The precipitation deficit is more intense in dense broadleaf forests, where it starts 2 years pre-fire and reaches -65 mm month$^{-1}$ before the fire event. SM anomalies (Fig. 6b) are similar for the three forest ecosystems. The SM deficit starts one year pre-fire and reaches -0.04 m$^3$ m$^{-3}$ before the fire event. Savannas and grasslands are affected later (6 months pre-fire) and to a lesser extent (-0.01 m$^3$ m$^{-3}$), as previously observed in Fig. 5. TWS anomalies are negative from 12 months pre-fire, excepted for grasslands. Again, dense broadleaf forests are the most impacted ecosystem, with a minimum TWS anomaly of -7 before fire. Temperature anomalies are not significant before 6 months pre-fire, excepted in grasslands and savannas where negative anomalies are visible. From 6 months pre-fire, temperature anomalies show a surplus in all ecosystems but savannas, and reach +0.9°C in needleleaf forests, +0.8°C in dense broadleaf forests, and +0.5°C in grasslands before the fire event. In summary, pre-fire drought is mainly observed in forests, with particularly low hydrological values in dense forests (rainforests), and particularly high temperatures in needleleaf forests (boreal ecosystems). Savannas and grasslands barely suffer from pre-fire drought. Temperatures are even mild one year pre-fire.

VVs anomalies were averaged within time frames of 6 months, from 1 to 36 months post-fire, in order to observe the global impacts and recovery time (Fig. 7). We considered that a variable has totally recovered when its anomaly is between -0.005 and +0.005. Immediately after fire, EVI is the most impacted variable, with average anomalies of -0.06 over broadleaf forests, -0.04 over needleleaf forests and savannas, and -0.03 over grasslands (Fig. 7a). EVI recovers rapidly, in about 25 to 30 months over needleleaf forests, and 19 to 24 months over other ecosystems. X-VOD is less affected over forests (-0.02) than over low vegetation (-0.03) (Fig. 7b). X-VOD gets back to normal within 3 years, needleleaf forests and savannas being the longest to recover. C-band is mainly impacted over forests and savannas, but barely over grasslands (Fig. 7c). C-VOD recovers slower than X-VOD, in particular over forests. L-VOD is mainly affected over broadleaf forests, and particularly over the densest ones (Fig. 7d). There, negative anomalies decrease up to -0.06 one year post-fire, then slowly increase. L-VOD is less affected than C-VOD elsewhere. It also shows a delayed impact by one year over needleleaf forests, as for C- and X-VOD.



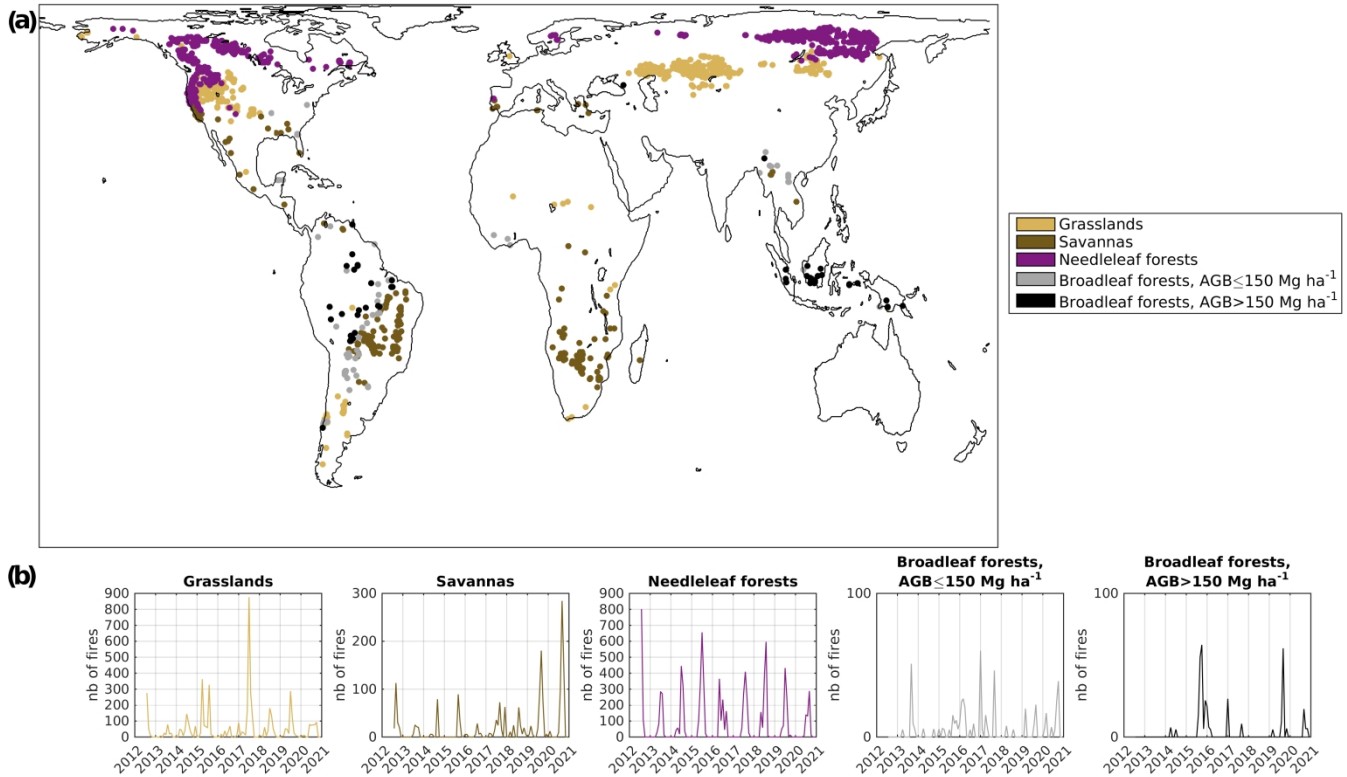

**Figure 4 – (a) Location of the selected fires and (b) histograms of the fire dates, for grasslands (IGBP label 10), tropical and subtropical savannas (IGBP labels 8 and 9, latitude < 40°), needleleaf forests (IGBP labels 1 and 3), sparse broadleaf forests (IGBP labels 2 and 4, AGB ≤ 150 Mg ha$^{-1}$), and dense broadleaf forests (IGBP labels 2 and 4, AGB > 150 Mg ha$^{-1}$). Australia was excluded as well as snow covered areas and seasonally flooded areas (see Sect. 3.2).**







**Figure 5 – EVI, X-, C-, L-VOD, P, SM, TWS, and T anomalies time series, shifted on the fire date, for (a) 520 points in the grassland biome; (b) 232 points in the savanna biome; (c) 701 points in the needleleaf forest biome; (d) 69 points in the sparse broadleaf forest biome; and (e) 48 points in the dense broadleaf forest biome. The missing values are mainly due to snow filtering.**

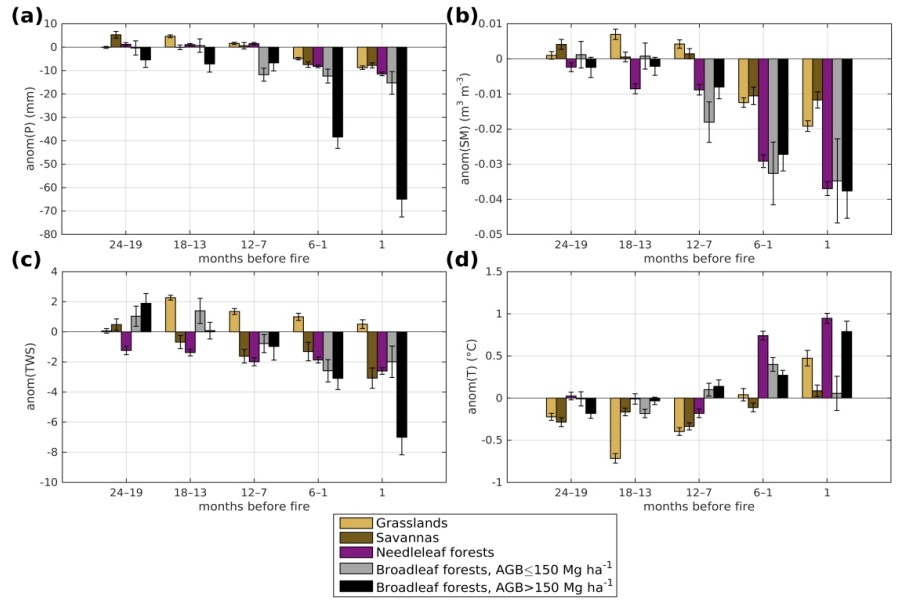

**Figure 6 – (a) precipitation, (b) SM, (c) TWS, and (d) temperature anomalies, for each land cover class, at several pre-fire time-scales. The error bars were computed with Eq. (3).**

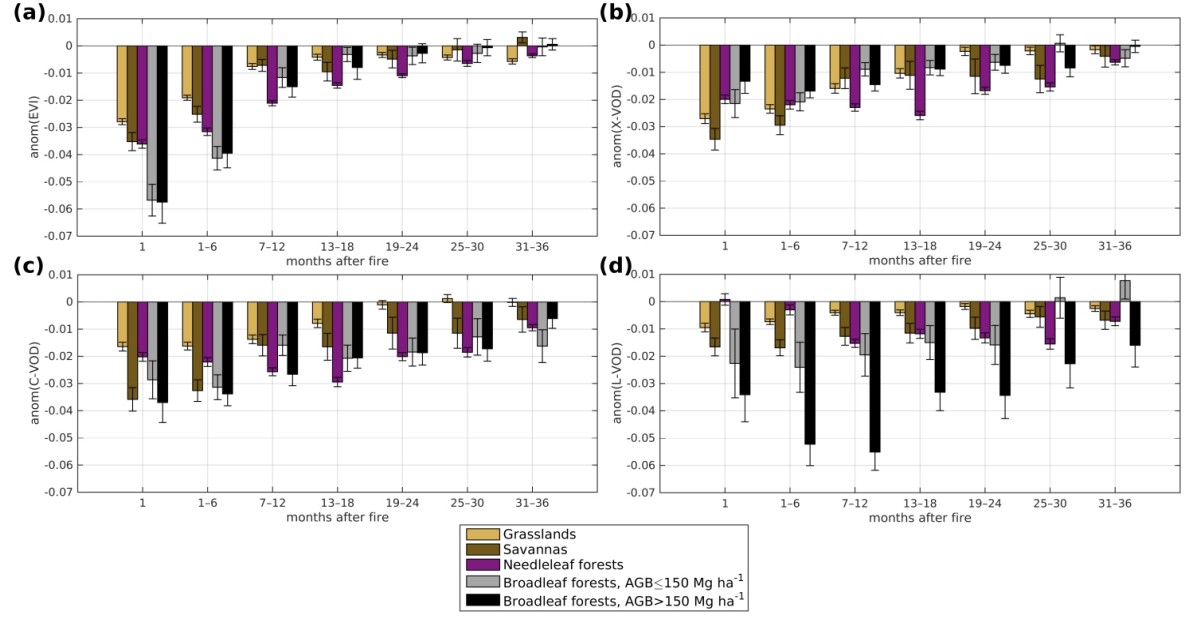

**Figure 7 – (a) EVI, (b) X-VOD, (c) C-VOD, and (d) L-VOD anomalies, for each land cover class, at several post-fire time-scales. The error bars were computed with Eq. (3).**





## 5. Discussion

### 380   5.1. Case study : analysis of three major fires

In Australia (fig. 3a), Ehsani et al. (2020) explained that the air temperature from December 2019 to February 2020 was about 1°C higher than usual, which increased evapotranspiration. The pre-fire lack of precipitation, due to a high-pressure system over South-East Australia, prevented the soil from supplying the moisture demand, and led to a significant vegetation drying (fuel) that facilitated the propagation of fires. L-VOD

regained its pre-fire values within a year, meaning that the woody biomass was not entirely destroyed. Indeed, these eucalyptus forests are known to be somewhat fire resistant (Wilkinson and Jennings 1993; Caccamo et al., 2015). They can regenerate branches and leaves by resprouting from heat-resistant buds (Burrows, 2002). The rapid recovery of VVs could also be explained by an increase in vegetation water content, linked with the post-fire increase in precipitation and SM (Konings et al., 2021). In 2020, SM values exceeded those of the

previous decade (anom(SM) = +0.15 $m^3$ $m^{-3}$), corresponding to the end of the severe drought affecting South-East Australia associated with the 2020/2021 La Niña event (Australian Bureau of Meteorology). The increase of precipitation may also have expedited the extinction of fires (Ehsani et al., 2020).

In California, all climate and fuel conditions leading to the 2018 widespread Mendocino complex described in Brown et al. (2020) are visible in Fig. 3b. We can observe positive rainfall and SM anomalies in winter

2016/2017, which led to the second consecutive spring with above average accumulation of fine fuel (grasses). Then, we can observe positive temperature anomalies in winter 2017/2018, when a lack of storm enabled the survival of grasses. In April 2018, precipitation and warm temperatures led to above normal spring brush and grass growth. No major drought is visible in summer 2018, but low rainfall and warm temperatures led to a rapid drying of fuels, and induced a poor overnight humidity recovery. The dramatic fire impacted EVI and L-

VOD in the long term. Eucalyptus, pine trees and chapparal were burnt. Even if this type of vegetation is fire-adapted, the severity of the fire seemed to have destroyed most of it. Increased forest fire activity in recent decades has likely been enabled by the legacy of fire suppression, human settlement, and anthropogenic climate change (Abatzoglou and Williams, 2016). Stephens et al. (2018) stated that the massive current tree mortality in California will undoubtedly provoke severe "mass fires" in the coming decades, driven by the

amount of dry and combustible wood.

The wildfire in Santarem region (Fig. 3c) was attributed to high temperature and low precipitation linked with El Niño event in winter 2015 (Berenguer et al., 2018). These extreme drought conditions worsened during the fire, and may explain its severity. The 3-year recovery time of EVI after the severe fire indicates a moderate regrowth of leaves and grasses. In contrast, L-VOD never regained its pre-fire values, meaning that trunks

were impacted on the long term.





## 5.2. Extension to the global scale

At global scale, grasslands, tropical and subtropical savannas do not show signs of pre-fire drought (Fig. 5a, 5b, 6). A substantial increase in VVs occurs 1–2 years before fire, which implies an increase in the available fuel, and may contribute to trigger the wildfires. Indeed, the fire risk in savannas is highest for a dry vegetation with enough fuel to enable a drastic burning (Mbow et al., 2004). This vegetation growth might be enabled by negative pre-fire temperature anomalies, and light positive pre-fire hydrological variables anomalies (Fig. 6). VVs are less impacted by fires (Fig. 7) because they are rapid and burn through the grass layer, resulting in less destruction than in forests (Menaut, 1983; Bessie and Johnson, 1995; Gignoux et al., 1997). L-VOD in particular is slightly impacted because the burnt vegetation is mainly leaf biomass. EVI quickly recovers after fire, probably because fire burns most of the AGB of grass species, but spares their large underground root systems, resulting in a rapid establishment of new shoots (Hochberg et al., 1994). The extremely high pre-fire VVs values are never regained.

In needleleaf forests time series (Fig. 5c), the numerous missing values correspond to the filtering of snow in winter. Wildfires in this biome are located in the Northern hemisphere temperate/boreal forests (Fig. 4a), and mostly occur in late spring and in summer (Fig. 4b). De Groot et al. (2013) stated that most boreal fires in Canada are due to lightning strikes and occur during summer; whereas most fires in Russia are human-caused and occur during spring. We found a strong pre-fire drought in this ecosystem (low SM, high T, Fig. 6), which is well documented (Weber and Stocks, 1998). VODs also showed a light surplus before fire, possibly linked with litter thickening, which also increases the fire risk. Indeed, de Groot et al. (2013) explained that in Russian boreal forests, extensive areas of dead light surface fuels (e.g. dead needles, cured grass, leaf litter) are available in spring and facilitate fire propagation. We found a delayed impact of fire on VVs, and a longer recovery time than in other ecosystems, of about 3–4 years (Fig. 5c and 7). Based on NDVI, Goetz et al. (2006) found a 5 year recovery time in Canada. Jin et al. (2012) found that the summer EVI has recovered in 5 to 8 years after fire. These findings also confirm previous results from Yang et al. (2017), who showed with NDVI analyses over North America that the fire effect on high latitude needleleaf trees was stronger and longer than on other vegetation types. Indeed, in North America, boreal forest fires are predominantly stand-replacing and high-intensity crown fires (Stocks et al., 2004; Jin et al., 2012). In Eurasia, most fires occur as surface fire and are usually of low to moderate intensity (de Groot et al., 2013). We found that L-VOD is less impacted by fire than other VVs in this biome. It can be explained by a dominant destruction of canopy leaves and branches by crown fires (Alexander and Cruz, 2011).

Sparse broadleaf forests (AGB ≤ 150 Mg ha$^{-1}$) subject to wildfires are mostly located in subtropical and temperate areas of South America, North America, West Africa, and South-East Asia (Fig. 4a). A drying trend is visible 1 year pre-fire (Fig. 5d and 6). The link between drought and wildfires was previously observed by de





Marzo et al. (2021) in the Argentine Gran Chaco; by Cheng et al. (2013) in the Mexican Yucatan forest; and by
Vadrevu et al. (2019) in South-East Asia, with a prominent influence of precipitation variations over
temperature variations. L-VOD and EVI are particularly impacted by fire, but they recover quickly (1 year for
EVI, 2 years for L-VOD). Yang et al. (2017) also found a rapid recovery time over North American broadleaf
trees due to their fire-adaptive resprouting regeneration mode. Same observations were made in the fire-
prone Argentine Chaco forest by Torres et al. (2014).

Dense broadleaf forests are mostly located in the tropics (Fig. 4a). We can notice few fires in the densest
rainforests (Congo basin, central Amazon) as they are usually too humid to burn (Cochrane, 2003); and
because seasonally flooded areas were excluded. A consistent drought is visible 8 months before fire (Fig. 5e).
We observed high negative SM, TWS, and precipitation anomalies in particular (Fig. 6). Chen et al. (2013) also
found TWS deficits before severe fire seasons across the southern Amazon. Drought-related fires were even
suggested to prevail over deforestation fires in the Amazon, and are predicted to increase in the near future
(Aragão et al., 2018). Indeed, rainfall shortage generates high water deficits (i.e. high negative SM anomalies),
which causes tree mortality, leaf shedding (visible in pre-fire EVI decrease) and thus increases fuel availability.
The opening of forest canopies also boosts incident radiation levels which leads to temperature rise (Ray et al.,
2005). The combination of fuel increase in a drier and hotter environment converts natural forests into fire-
prone ecosystems (Aragão et al., 2018). We also found that the dense broadleaf forest biome was the most
impacted by fire (Fig. 7), because it is not a fire-adapted ecosystem (Cochrane, 2003). L-VOD in particular
decreases strongly and recovers very slowly (Fig. 7d), as previously observed over Santarem fire (Fig. 3c). The
strong post-fire decrease in L-VOD is due to biomass destruction but also to water stress in the remaining
vegetation (Konings et al., 2021). This finding confirms the significant and damaging impact of fires in the
dense broadleaf ecosystem previously observed by Silva et al. (2018) and de Faria et al. (2021). L-VOD was
previously proven to be more sensitive to high AGB values than C- and X-VOD (Rodriguez-Fernandez et al.,
2018). Here, we suggest that L-VOD depicts better the fire impact on high AGB areas than the other VVs.

For all biomes, EVI is the most rapid index to recover, because leaves rapidly resprout. EVI and X-VOD seem
better adapted for grasslands fire monitoring, C-VOD for savanna fire monitoring, and L-VOD for forest fire
monitoring.

## 5.3. The potential of L-VOD for vegetation recovery monitoring over dense forests

VV anomalies were also plotted with respect to the number of fires in the dense broadleaf ecosystem,
immediately after fire (from 1 to 3 months post-fire, Fig. 8a) and over a longer period (from 1 to 2 years post-
fire, Fig. 8b). A quasi-linear relationship is visible for all VVs. As previously observed in Sect. 4.2, EVI and L-VOD
are the most impacted variables immediately after fire (Fig. 8a), while L-VOD is still significantly affected 1 to 2



years after fire (Fig. 8b). L-VOD then shows a clear response to fire events over high AGB areas, immediately after fire but also in the long term, and proportionally to the number of fires within a SMOS pixel. Thanks to its sensitivity to coarse woody elements and to its deep penetration through the vegetation layer, L-VOD is better correlated to high AGB than other VVs (Rodriguez-Fernandez et al., 2018), and could be used for post-fire
recovery monitoring over dense forests.

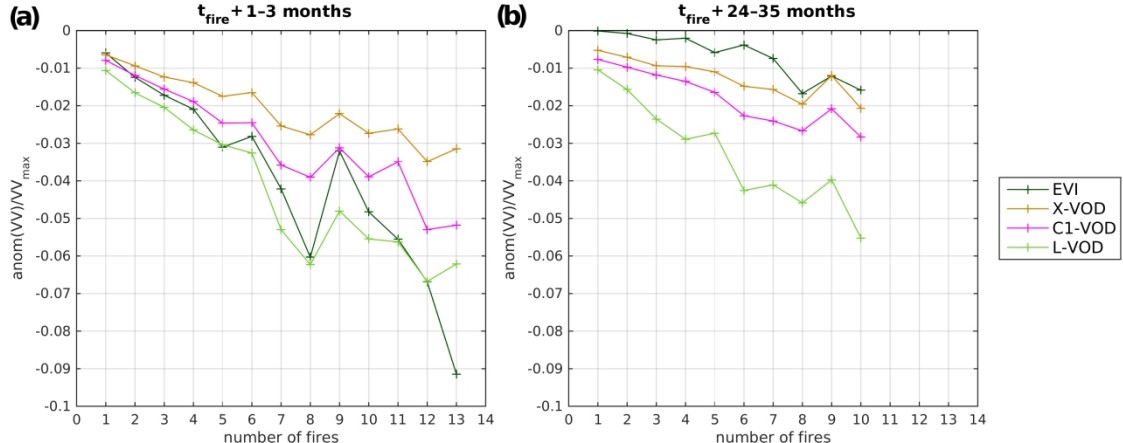

**Figure 8 – VV anomalies averaged (a) from 1 to 3 months post-fire, and (b) from 24 to 35 months post-fire, with respect to the number of fires by pixel (MODIS), for dense broadleaf forests only. VV anomalies were normalized using their respective 99th quantiles VV$_{max}$ (0.60 for EVI, 1.02 for**
**X-VOD, 1.20 for C-VOD, and 1.21 for L-VOD).**

## 6. Conclusion

In this paper, we analysed the pre-fire behaviour of four fire likelihood factors, including SMOS SM which provides access to top surface soil moisture at the global scale. In forests, which generally maintain a steady humidity, we found that fires are linked with intense and prolonged drought. Pre-fire temperature anomalies
are particularly high in boreal needleleaf forests. In savannas and grasslands, in agreement with previous studies (Mbow et al., 2004), we found evidences of an increase in available fuel prior to fire events, enabled by humid and cold conditions a few years before. We also found that vegetation variables recover rapidly in these ecosystems, as wildfires are often rapid and mildly destructive for trees. In contrast, over forests, fires can damage the vegetation in the long term. Zhang et al. (2021) demonstrated the potential of C-band vegetation
optical depth to detect the vegetation change patterns caused by fire in the southern Amazon. Our study confirms these findings and extends it to the global scale, and to two extra wavelengths. Dense broadleaf forest fires particularly impact the L-band emission, which represents coarse woody elements (trunks and stems); whereas sparse vegetation fires affect more C- and X-bands, which are more sensitive to small branches and leaves. For all biomes, the optical-based index (EVI) drops down after fire but recovers quickly,



as it represents only herbage and canopy foliage. The long term impact on L-VOD in dense broadleaf forests shows that fires in this ecosystem are severely destructive for trunks, while smaller woody elements and leaves resprout faster. Thus, L-VOD seems the best adapted vegetation variable for the detection of fires and the monitoring of post-fire recovery over dense vegetation.

The increasing number of wildfires threatens the stability of several ecosystems. It is then particularly
important to monitor the vegetation health and L-band proved to be complementary to existing measurements, especially over dense forests.

**Author contributions**

E.B., A.M., N.J.R.F., and Y.H.K. planned the research discussed in this manuscript. E.B. performed most of the computations. S.M. provided the AGB dataset and expertise on AGB and on forest loss estimation. All authors
participated in the writing and provided comments and suggestions.

**Competing interests**

The authors declare that they have no conflict of interest.

**Acknowledgements**

E.B., A.M., N.J.R.F., Y.H.K. acknowledge support by CNES (Centre National d'Etudes Spatiales) TOSCA program.
SMOS L2 was obtained from ESA's DPGS (Data Processing Ground Segment). The authors would like to thank the European Space Agency (ESA). AMSR-2 data were provided by Vrije Universiteit Amsterdam (Richard de Jeu) and NASA GSFC (Manfred Owe) (2014), AMSR2/GCOM-W1 surface soil moisture (LPRM) L3 1 day 25 km × 25 km ascending V001, Edited by Goddard Earth Sciences Data and Information Services Center (GES DISC) (Bill Teng), Greenbelt, MD, USA, Goddard Earth Sciences Data and Information Services Center (GES DISC),
Accessed: 29/07/2020, doi:10.5067/M5DTR2QUYLS2. TWS data were obtained from GravIS web portal, the Gravity Information Service of the German Research Centre for Geosciences (GFZ). Fires, temperature and EVI data came from NASA Earth Observations (NEO) portal and Land Processes Distributed Active Archive Center (LP DAAC). IMS Daily Northern Hemisphere Snow and Ice Analysis at 4 km resolution, Version 1, came from the United States National Ice Center (USNIC), Boulder, Colorado USA, delivered by the National Snow and Ice
Data Center (NSIDC), doi: https://doi.org/10.7265/N52R3P, Accessed: 01/08/2020.





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
