# Peer review of "Monitoring post-fire recovery of various vegetation biomes using multi-wavelength satellite remote sensing"

_Biogeosciences, 2021_

## Referee Comment (RC1)

**Review of the manuscript "SMOS L-VOD shows that post-fire recovery of dense vegetation is slower than what is depicted with X- and C-VOD and optical indices"**

This paper studies the time evolution of several climate and vegetation variables before (triggering factors) and after (recovery of vegetation) fire occurrences worldwide. The study is divided in two parts. The first part details fire episodes in the Amazon, California and Australia. The second part extends the research to a global scale. The authors confirm the capacity of different Earth observation sensors to capture drought situations leading to fire ignitions, and nicely show how vegetation recovery can be monitored with microwave and optical-infrared data. Importantly, they demonstrate which VOD frequencies are appropriate for monitoring vegetation recovery after fires in different land cover types. The main finding is that L-VOD, which is more related to tropical biomass, shows delayed recovery if compared to higher VOD frequencies and optical-infrared indices in this forest type.

The paper is well written and, as explained above, the findings are sounding. However, I have some major concerns that must be addressed before being accepted for publication. The most important one refers to the completeness of the fires database. Both major and minor comments are detailed hereafter.

**Major comments**

1. Figure 4 shows the fires studied in this work during a nine-years study period (July 2012 – December 2020). The authors explain that "the considered fires are well spread spatially […]." However, the map of fires is certainly omitting a large amount of wildfire episodes worldwide and, most importantly, it scarcely includes fire episodes for all relevant fire-prone regions. Probably the most relevant cases in that sense are the Sahel and the Mediterranean, where a large number of wildfires occur within the land cover types under study (grasslands, savannahs…), according to the monthly maps of the product applied. It is likely that, in part, these regions are not well represented in the study because it does not include shrubland covers. This land cover type should be included as well in this research. Hence, please ensure completeness for all fire-prone regions, especially the Mediterranean and the Sahel, and all land cover types (shrublands are lacking). With this, large and continuous fire occurrence patches (similar to those in the Russian and North American grasslands and forests) should be observed in the northern Mediterranean (especially southern Italy, the Iberian Peninsula and Greece), and in the Sahel.

   In the case of Australia, the authors appropriately excluded this continent as explained in section 3.2. However, justification for this exclusion is provided for vegetation variables and vegetation recovery. However, the authors should include the region at least for CVs explaining wildfires ignition in the region (i.e., SM, TWS and P).

   Also, it is quite surprising to me that the number of fires in tropical forests is very low. This is worrying as it can affect the representativeness of the results in tropical forest fires, and consequently the main conclusion of the paper (that L-VOD is the most appropriate for studying fire recovery in the tropics). Can the authors double-check that all fire occurrences in this region have been included?

2. Although the main focus of the paper is on vegetation recovery, the work also details which main climate and vegetation variables act as triggers of fire ignition (mainly precipitation, soil moisture, ground water storage, and fuel availability). In that sense, the introduction should be extended to provide further state of the art. On the one hand, GRACE data (groundwater storage) has been previously applied for fire risk assessment in the United States (e.g., Jensen et al., 2018; Farahmand et al., 2020). On the other hand, SMOS soil moisture data has been applied as an alternative source of moisture information in the McArthur Forest Fire Danger Index (FFDI; Holgate et al., 2017). Also, SMOS SM anomalies have been found to explain anomalous fire episodes in the northwestern Iberian Peninsula (Chaparro et al., 2016) and in Canada (Ambadan et al., 2020). A part from L-band, a nice study of how satellite soil moisture anomalies can be used for fire risk assessment is shown by Forkel et al. (2012; see also my minor comment below).

**Minor comments**

Line 15 (and through the entire paper): optical vegetation indices → optical-infrared vegetation indices. Or VIS/IR vegetation indices, if you prefer. The point is that EVI includes both visible and infrared bands.

L. 30: Amazônia legal → Amazônia Legal

L. 50: a sentence should be included about the fact that most wildfires are ignited due to human activities. In the Mediterranean regions 95% of fires are due to these causes, and similar percentages are found in other areas (e.g., 90% in South Asia, 85% in South America, 80% in Northern Asia; FAO, 2006).

L. 90-91: according to these lines, it seems that soil moisture could be retrieved only from L-band sensors, while this is not true. I suggest explaining the advantage of L-band (more penetration capacity through soils and vegetation) to provide better motivation on the advantage of using L-band for soil moisture retrievals, and to explain why L-band is more linked to dense biomass (this point is important for the interpretation of results in this paper).

L. 94-95: "This study also presents for the first time L-band used in conjunction with other sensors, from optical (EVI) to X- and C-band…": add (specify): "in the study of vegetation recovery after fires."

L. 120: from SMOS satellite → from the SMOS satellite.

L. 136: please specify which months are not included, and how much months does it add up within the entire study period.

L. 197: watern → water

L. 216: why are VOD data resampled to 1 km resolution and later averaged to the SMOS grid? This does not make sense because VOD at C- and X-bands have much coarser resolutions than 1 km (as in SMOS). Please, be sure to interpolate directly C- and X-VOD data from their native resolution to the SMOS grid. An intermediate step through 1 km may introduce errors.

L. 239: you use "ha" as burned area unit here, but "km$^2$" throughout the manuscript. Please be consistent, use only one or the other.

L. 241: how was burn severity defined and classified in "moderate", "high", etc… in this case?

L. 303-304: "a strong decrease during the fire event" → Also before it.

L. 311-312: it should be noted that the positive T anomalies and the negative TWS and P anomalies reach their maximum and minimum (respectively) at the end of the fire period. Can you provide a possible interpretation for this?

L. 348: please mention that savannahs and grasslands show positive VV anomalies one year before (as you will discuss it later in the discussion).

Figure 5: there is an interesting result in Fig. 5 which could be highlighted. Note that, in boreal forests, SM and TWS anomalies are negative also one year before fires. This is interesting as it could be in line with results shown in Forkel et al. (2012). In that case, the authors found that negative SM anomalies in Siberia during summer 2002 led to low amount of water being frozen within permafrost soils during winter 2002-2003. Therefore, a low amount of water was stored (frozen) and then released to the soils during permafrost melting in spring-summer 2003. This led to drier than usual soils in summer 2003, which eased the outbreak of large wildfires. In particular, the Forkel et al. stress in the abstract that "analyses of satellite data for 2002–2009 indicate that previous-summer surface moisture is a better predictor for burned area than precipitation anomalies or fire weather indices for larch forests with continuous permafrost." Your results are in line with this finding and this could be briefly included in the manuscript.

L. 340-345: when you comment on TWS and T anomalies, please refer to Figs. 6c and 6d, respectively.

L. 391: the reference to the Australian Bureau of Meteorology should be accompanied by a year and an appropriate reference within the reference list.

L. 401: can you quantify the severity of the fire? Actually, it would be interesting to mention severity indices when discussing the three study cases, if possible.

L. 428: "which is well documented" → "which is well documented in previous fire episodes in this region."

L. 444 and 449: Argentine → Argentina

**References**

Ambadan, J. T., Oja, M., Gedalof, Z. E., & Berg, A. A. (2020). Satellite-Observed Soil Moisture as an Indicator of Wildfire Risk. Remote Sensing, 12(10), 1543.

Chaparro, D., Piles, M., Vall-Llossera, M., & Camps, A. (2016). Surface moisture and temperature trends anticipate drought conditions linked to wildfire activity in the Iberian Peninsula. European Journal of Remote Sensing, 49(1), 955-971.

FAO, 2006. Fire Management – Global Assessment 2006. A Thematic Study Prepared in the Framework of the Global Forest Resources Assessment 2005. Food and Agriculture Organization, Rome.

Farahmand, A., Stavros, E. N., Reager, J. T., Behrangi, A., Randerson, J. T., & Quayle, B. (2020). Satellite hydrology observations as operational indicators of forecasted fire danger across the contiguous United States. Natural Hazards and Earth System Sciences, 20(4), 1097-1106.

Forkel, M., Thonicke, K., Beer, C., Cramer, W., Bartalev, S., & Schmullius, C. (2012). Extreme fire events are related to previous-year surface moisture conditions in permafrost-underlain larch forests of Siberia. Environmental Research Letters, 7(4), 044021.

Holgate, C. M., van Dijk, A. I., Cary, G. J., & Yebra, M. (2017). Using alternative soil moisture estimates in the McArthur Forest Fire Danger Index. International Journal of Wildland Fire, 26(9), 806-819.

Jensen, D., Reager, J. T., Zajic, B., Rousseau, N., Rodell, M., & Hinkley, E. (2018). The sensitivity of US wildfire occurrence to pre-season soil moisture conditions across ecosystems. Environmental research letters, 13(1), 014021.

---

## Author Comment (AC1)

**Dear Dr Chaparro,**

We want to thank you for your thorough proofreading and your relevant comments. We took them into account and believe the manuscript has been substantially improved thanks to your suggestions.

The major modification was the addition of the land covers "shrublands", "croplands", and "natural vegetation mosaic" to our study. The maximum threshold on the seasonal water fraction has also been increased from 10% to 20%. The exclusion of regions subject to seasonal fires and of Australia was better justified. The suggested references have been included. Other minor changes were made according to your suggestions, in order to improve the clarity of the text.

Please find below in blue font a detailed description of how we addressed your comments. Sincerely,

Emma Bousquet et al.

**Review of the manuscript "SMOS L-VOD shows that post-fire recovery of dense vegetation is slower than what is depicted with X- and C-VOD and optical indices"**

This paper studies the time evolution of several climate and vegetation variables before (triggering factors) and after (recovery of vegetation) fire occurrences worldwide. The study is divided in two parts. The first part details fire episodes in the Amazon, California and Australia. The second part extends the research to a global scale. The authors confirm the capacity of different Earth observation sensors to capture drought situations leading to fire ignitions, and nicely show how vegetation recovery can be monitored with microwave and optical-infrared data. Importantly, they demonstrate which VOD frequencies are appropriate for monitoring vegetation recovery after fires in different land cover types. The main finding is that L-VOD, which is more related to tropical biomass, shows delayed recovery if compared to higher VOD frequencies and optical-infrared indices in this forest type.

The paper is well written and, as explained above, the findings are sounding. However, I have some major concerns that must be addressed before being accepted for publication. The most important one refers to the completeness of the fires database. Both major and minor comments are detailed hereafter.

**Major comments**

1. Figure 4 shows the fires studied in this work during a nine-years study period (July 2012– December 2020). The authors explain that "the considered fires are well spread spatially [...]." However, the map of fires is certainly omitting a large amount of wildfire episodes worldwide and, most importantly, it scarcely includes fire episodes for all relevant fire-prone regions. Probably the most relevant cases in that sense are the Sahel and the Mediterranean, where a large number of wildfires occur within the land cover types under study (grasslands, savannahs...), according to the monthly maps of the product applied. It is likely that, in part, these regions are not well represented in the study because it does not include shrubland covers. This land cover type should be included as well in this research. Hence, please ensure completeness for all fire-prone regions, especially the Mediterranean and the Sahel, and all land cover types (shrublands are lacking). With this, large and continuous fire occurrence patches (similar to those in the Russian and North American grasslands and forests) should be observed in the northern Mediterranean (especially southern Italy, the Iberian Peninsula and Greece), and in the Sahel.

The reviewer is right, we omitted the land cover class "shrublands" to focus only on five biomes to lighten the observations and not to disperse our efforts. Nevertheless, taking into account the referee's comments, we decided to add this land cover (IGBP labels 6 and 7) to the biome "tropical and subtropical savannas" (IGBP labels 8 and 9). We also added the land covers "croplands" (IGBP label 12) and "natural vegetation mosaic" (IGBP label 14) to the biome "grasslands" (IGBP label 10). Figure 4 of the manuscript was replaced by the resulting Fig. R1 below.

Despite this modification, the number of points is still low in the Sahel, because of the selection method described at lines 280-282 : "Fires with a number larger than 5 in the MODIS dataset were considered, and only if no other fire occurred on the same area (number lower than 2 apart from the main fire event). This was done to observe only the impact of the major fires, without any other disturbance." In the Sahel, these

conditions are not fulfilled because many fires occur each year, and the second threshold is exceeded. An example is shown in Fig. R2.

The above paragraph was replaced by : "To properly observe the factors and impacts of a fire event without any other disturbance, only 25 km regions showing a unique and heavy fire over the time period were considered. This excluded areas with regular seasonal fires, such as the Sahel region. For that, a minimum threshold of 5 was applied on the maximum number of fires ; and a maximum threshold of 2 was applied outside the main fire event period (i.e. outside the period -6/+6 months around the fire event)."

Figure R1 - Location of the selected fires and histograms of the fire dates, for grasslands and croplands (IGBP label 10, 12 and 14), savannas and shrublands (IGBP labels 6, 7, 8 and 9), needleleaf forests (IGBP labels 1 and 3), sparse broadleaf forests (IGBP labels 2 and 4, AGB  $\leq$  150 Mg ha-1), and dense broadleaf forests (IGBP labels 2 and 4, AGB  $\geq$  150 Mg ha-1). Australia was excluded as well as areas affected by water, snow, or strong topography.

---

## Author Comment (AC2)

Dear Dr Forkel,
We wish to thank you for your thorough proofreading and your relevant comments. They were taken into account in order to improve the manuscript.
In that respect, the contributions of VWC and AGB in VOD were further discussed. The exclusion of Australia and of regions subject to seasonal fires was also better justified. The suggested references have been included. Other minor changes were made according to your suggestions, in order to improve the clarity of the text and of the figures.
Please find below in blue font a detailed description of how we addressed your comments.
Sincerely,
Emma Bousquet et al.

**RC2**
**Review of the manuscript "SMOS L-VOD shows that post-fire recovery of dense forests is slower than what is depicted with X- and C-VOD and optical indices" by Bousquet et al. 2021**

The study investigates the temporal anomalies of different remotely-sensed climate variables and vegetation properties before and after fire. The main finding is that L-VOD recovery slower after fire than X-VOD, C-VOD or optical vegetation indices. The study provides an important contribution to the research on the understanding of VOD data in different wavelengths and on the use of VOD to study fire. The text is written in an easy to understand language, however some sentences are written too simple and are unspecific and by this make partly wrong statements (see specific comments).

Fire is both driven by fuel loads and fuel moisture (Chuvieco et al., 2014) and VOD is sensitive both to biomass and fuel moisture content (FMC), or vegetation water content (VWC) (Konings et al., 2019). Especially, X-VOD is more sensitive to FMC than other microwave-derived surface properties (Fan et al., 2018). Hence VOD was already successfully used in data-driven fire models (Forkel et al., 2017; Kuhn-Régnier et al., 2021). However, it is not yet clear how the different VOD bands are sensitive to either biomass or FMC and hence of the biomass- or moisture-"component" of VOD is more important for fire prediction. The same question is true for post-fire recovery: Does VOD measure really a recovery of biomass or a recovery of the fuel moisture? Note that different changes in VOD and optical indices and their relation with land cover trends have been also investigated in Andela et al. (2013). Those sensitivities of VOD to biomass and moisture are important to consider in the context of this study, however, those lines of thought were not included in the introduction and/or discussion. Hence, the objective of the study is also not entirely clear. Are you aiming to use VOD as a proxy for post-fire biomass recovery? If yes, how could you disentangle the moisture effect? The role of biomass and moisture on VOD should be introduced in chapter 1 and discussed in the discussion.

This paper aims at studying post-fire vegetation recovery through four variables, EVI, X-VOD, C-VOD, and L-VOD. We cannot study biomass recovery alone because EVI represents only the greenness of vegetation, and VOD is indeed sensitive to both biomass (AGB) and vegetation water content (VWC). This was stated at line 144 and briefly discussed at lines 388 for Australia and at lines 463 for the dense broadleaf ecosystem. Disentangling the contributions of AGB and VWC in the VOD is very tricky and has not been done yet, because they are strongly linked (Konings et al., 2019). We can only provide assumptions. For example, when VOD and SM evolutions follow each other, we can assume that the fuel moisture (VWC) is impacted. Monitoring directly AGB is complex since monthly AGB maps do not exist. In the first part, we considered the forest loss from Hansen et al. (2013) "lossyear" product to corroborate that fires lead to biomass destruction, and not only to vegetation drying. We agree to develop the discussion on that matter though, thanks to the relevant references you provided us. For that, we added (modifications appear in red):
L. 72 : "*Indirectly, drought also causes leaf shedding, branch losses, and a decrease in fuel moisture content, which increases forest flammability (Nepstad et al., 2001; Pausas and Bradstock, 2007; Chuvieco et al., 2014). Surveying the soil moisture (SM) and the vegetation water content (VWC) could then be a good indicator for fire risk detection. [...] AMSR-E VOD was successfully used in data-driven fire models (Forkel et al., 2017; Kuhn-Régnier et al., 2021).*"
L. 99 : "*Microwave VODs are also sensitive to VWC and can help to monitor the biomass status (Fan et al., 2018; Konings et al., 2019).*"

*L. 146 : "No clear approach exists for disentangling the contributions of AGB and VWC in the VOD because of the co-sensitivity of microwave observables to both quantities (Konings et al., 2019)."*

*L. 381 : "In South-East Australia, a strong pre-fire drought is visible in the climate variables but also in the mild decrease of vegetation variables (Fig. 3a), linked with VWC deficit."*

*L. 413 : "A substantial increase in VVs occurs 1–2 years before fire, which implies an increase in vegetation density, e.g. available fuel. Immediately before fire, both VOD and SM values drop down, suggesting a decrease in VWC, especially over grasslands (Fig. 5a). The increase of vegetation material combined with the decrease of VWC may contribute to trigger large wildfires (Forkel et al., 2017; Kuhn-Régnier et al., 2021)."*

*L. 456 : "Indeed, rainfall shortage generates high water deficits (i.e. high negative TWS and SM anomalies), which cause tree mortality, leaf shedding (visible in pre-fire EVI decrease) and thus increase fuel availability (Aragão et al., 2018). Nevertheless, no pre-fire VOD decrease is observed here, showing that tree species of dense forests can maintain their VWC."*

*L. 478 : "Thanks to its sensitivity to coarse woody elements and to its deep penetration through the vegetation layer, L-VOD is better correlated to high AGB than other VVs (Rodriguez-Fernandez et al., 2018), and could be used for post-fire recovery monitoring over dense forests. One must keep in mind that not only the biomass volume (AGB) but also the biomass status (VWC) is depicted in VOD."*

Like the second referee, I am puzzled about the selection of fires for the analysis (Fig. 4a). Globally, fire activity is strongest in the African Savannahs (mostly in the Sahel), in northern Australia and South America (Giglio et al., 2013; Andela et al., 2019). However, the Sahel and Australia (and the Mediterranean) are almost completely missing in the selection of fires. What are the reasons? Does it make sense to not consider some of the most fire-prone regions in the analysis? You need to revise this (or clarify it).

The Sahel and the Mediterranean are missing because of the selection method chosen : *"Fires with a number larger than 5 in the MODIS dataset were considered, and only if no other fire occurred on the same area (number lower than 2 apart from the main fire event). This was done to observe only the impact of the major fires, without any other disturbance"* (lines 280-282). In the Sahel, these conditions are not fulfilled because seasonal fires occur and the second threshold is exceeded (Fig. R1).

The above paragraph was unclear and was replaced by : *"To properly observe the factors and impacts of a fire event without any other disturbance, only 25 km regions showing a unique and heavy fire over the time period were considered. This excluded areas with regular seasonal fires, such as the Sahel region. For that, a minimum threshold of 5 was applied on the maximum number of fires ; and a maximum threshold of 2 was applied outside the main fire event period (i.e. outside the period -6/+6 months around the fire event)."*

We also added the land cover "shrublands" (IGBP labels 6 and 7) to the biome "tropical and subtropical savannas" (IGBP labels 8 and 9); and "croplands" (IGBP label 12) and "natural vegetation mosaic" (IGBP label 14) to the biome "grasslands" (IGBP label 10). Figure 4 of the manuscript was replaced by Fig. R2 below.

As for Australia, by using the methodology described in Sect. 3.2, the majority of fires in Australia occurred in 2012 in the Outback (shrublands) and in 2019/2020 in the South-East (broadleaf forests). These two cases correspond respectively to the very beginning and the end of the study period. This prevents a robust pre- and post-fire study, especially since these fires become predominant in the global dataset (Fig. R3): Australia represents 57% of the total fires for the savannas and shrublands biome, and 54% for the dense broadleaf forests biome. This continent is strongly over-represented in the MODIS active fire product due to the large size of the fire events over this continent (Giglio et al., 2016). Thus, we reckon that it is more consistent not to analyse those events in the current study. We will certainly revisit the study in Australia with more hindsight in the future, once the time series after the fire will be long enough to properly analyse the vegetation recovery. The text was modified as follows to better justify this exclusion: *"Australia was excluded because numerous fires were detected in 2012 in the Outback (shrublands) and in 2019/2020 in the South-East (broadleaf forests), which prevailed over the global dataset (~55% of the points) and prevented to perform a robust pre- and post-fire study."*

In order not to omit this continent from the study, we included a detailed case study in South-East Australia in the first part (Fig. 3a).

[Figure]

Figure R1 - Time series of the number of fires (MODIS active fire product) over a 25 km pixel in the Sahel (red). The blue line represents the minimum threshold s1 = 5 for the strongest fire event detection; and the green line represents the maximum threshold s2 = 2 for the rest of the period (which is exceeded).

Figure R2 - Location of the selected fires and histograms of the fire dates, for grasslands and croplands (IGBP labels 10, 12, and 14), savannas and shrublands (IGBP labels 6, 7, 8, and 9), needleleaf forests (IGBP labels 1 and 3), sparse broadleaf forests (IGBP labels 2 and 4, AGB ≤ 150 Mg ha$^{-1}$), and dense broadleaf forests (IGBP labels 2 and 4, AGB > 150 Mg ha$^{-1}$). Australia was excluded as well as areas affected by water, snow, or strong topography (see Sect. 3.1).

[Figure]

Figure R3 - Same as Fig. R2, without the exclusion of Australia.

**Specific comments**

L 34: "Authors found" – unclear which authors are meant
It refers to the previous authors (Silva et al., 2018). The sentence was modified as follows: *"Specifically, they found [...]"*.

L 94-95: The sentence is not clear and wrong. L-band was already used in conjunction with other VOD data. See for example (Fan et al., 2018)
The reviewer is right. We modified this sentence by: *"This study also presents for the first time L-band used in conjunction with other sensors, from optical-infrared (EVI) to microwave X- and C-bands, for the study of post-fire vegetation recovery."*

L 96: Using the abbreviation "VV" for vegetation variables in a study that makes use of microwave remote sensing data is not ideal because VV stands for vertically polarized sensed and received radiation. In order to avoid confusion, I recommend using another abbreviation.
We do not think there is much room for confusion as radar is not used in this study. However, following the reviewer comment, we decided to remove the abbreviation "VV".

L 208-209: What is the size of those fires relative to the spatial resolution of the SMOS data? How many pixels are included in those fires? Did you apply any threshold for minimum fire sizes in order to select the case studies?
MODIS Active Fire product is a quantification of the number of fires observed within a 1000 km² area. A fire must cover at least ~ 1000 m² to be detected (see Sect. 2.1). The thresholds applied to the MODIS number of fires for the selection of the studied regions are described at lines 280-282. We added *"(see Sect. 3.2)"* to

facilitate the understanding. For the three case studies, the selection was arbitrary and not based on the MODIS number of fires. We added the description of the number of fires observed in Fig. 3 in Sect. 4.1:
*"In evergreen forests of the South Coast of New South Wales in Australia (Fig. 3a), fires reach a maximum in January 2020 (mean number of fires = 8)."*
*"The Mendocino Complex was the strongest of the three case studies, with 20 fires observed in average in August 2018."*
*"In the dense rainforest near Santarem (Brazilian Amazon), only 4 fires were detected in December 2015 (Fig. 3c), but this value may be underestimated due to cloud coverage (Giglio et al., 2020)."*

L 256: Here a caption for a sub-chapter is missing because the chapter is still about the Santarem case study and not about the data analysis.
The reviewer is right, we added a subsection *"3.1.4 Time series anomalies computation"*.

L 274: I don't understand the point iii – You excluded entire Australia from the global analysis? Why? Can you improve this description.
We agree that the justification of the exclusion of Australia was not clear enough. We modified this paragraph accordingly (see the answer to your second major comment).

L 280-285: How was the anomaly computation done if a pixel experienced multiple fires?
Pixels experiencing multiple fires were removed because we wanted to observe the factors and impacts of a fire event without any other disturbance (see the answer to your second major comment).

L 413: Those time lags between fire and fuel availability were also recently reported here (Kuhn-Régnier et al., 2021).
Thank you for this reference. It was added in the text (see the answer to your first major comment).

L 419: What does L-VOD "measure" in grasslands and savannahs? L-VOD should be close to zero in grasslands as the L-band microwave penetrate the grass layer. Can you provide some more explanation and references about the sensitivity of L-VOD in grasslands?
Over grasslands, L-VOD is indeed very small but not negligible. Actually, the VOD is never zero when there is vegetation. SMOS L-VOD = 0.22 on average over grasslands, which is quite low (L-VOD spans from 0 in deserts to ~1.3 in dense forests). The effects of grasslands, fallow and crops have been reported in many studies (Wigneron et al., 2004; Wigneron et al., 2012; Togliatti et al., 2019; Togliatti et al., 2021). L-band penetrates through the grass layer but is still attenuated by it, in particular when its water content is significant (Saleh et al., 2006). Moreover, some grasslands are made of a thick grass layer which can reach one metre or more, and thus are not transparent for L-band. Panciera et al. (2011) characterised grasslands and croplands as "moderate" canopy conditions.
In savannas and woody savannas, SMOS L-VOD = 0.33 and 0.49 respectively. Savannas are described as "lands with herbaceous and other understory systems, and with forest canopy cover between 10−30%" (Broxton et al., 2014). The sparse trees of this mixed ecosystem contribute predominantly to the L-VOD signal.

L 425-426: Note that there are much more differences in fire regimes between North American and Siberian boreal forests (Rogers et al., 2015). In your analysis, different signals might be mixed and hence it would be better to investigate the two regions separately.
Thank you for this useful reference. It was added in the manuscript. Rogers et al. (2015) explained that fires in North America are predominantly high intensity crown fires, whereas fires in Eurasia are generally lower intensity surface fires, which are less destructive for the vegetation. This was already written in the discussion (lines 436-438): "in North America, boreal forest fires are predominantly stand-replacing and high-intensity crown fires (Stocks et al., 2004; Jin et al., 2012). In Eurasia, most fires occur as surface fire and are usually of low to moderate intensity (de Groot et al., 2013)."
Rogers et al. (2015) also stated that these different fire regimes are due to different tree species.
We plotted separately the time series on the two regions (Fig. R4). We can observe some similarities: i) a pre-fire surplus in vegetation variables; ii) high temperature and low hydrologic variables during fire; iii) a

strong impact of fire on VVs, mainly EVI; iv) a long recovery phase. Some discrepancies are also visible: in North America, i) pre-fire SM and TWS deficits are higher, while T surplus is lower; ii) EVI and L-VOD are more impacted by fire, while C- and X-VOD are less impacted; iii) L-VOD and TWS continue to decrease two years after fire; iv) EVI and L-VOD recover slower than in Eurasia (~ 3 years vs ~ 2 years for EVI, ~ 4 years vs ~ 2 years for L-VOD). The latter observation can be due to the fact that, as the reviewer states, North American crown fires are more destructive for the vegetation than Siberian surface fires.

These time series (Fig. R4) will be provided in Supplementary Material, and the associated observations added in the discussion: *"Fires in North America are predominantly stand-replacing and high-intensity crown fires (Stocks et al., 2004; Jin et al., 2012), whereas fires in Eurasia are generally lower intensity surface fires, less destructive for the vegetation (de Groot et al., 2013). These different fire regimes are influenced by tree species (Rogers et al., 2015). Time series were plotted separately over each continent (Fig. S1). L-VOD and EVI recover slower in North-America than in Eurasia (~ 4 years vs ~ 2 years for L-VOD, ~ 3 years vs ~ 2 years for EVI), confirming these different boreal fire regimes."*

[Figure]

Figure R4 - Time series of the number of fires, and anomalies time series of EVI, X-, C-, L-VOD, P, SM, TWS, and T, shifted on the fire date, for a) American needleleaf forests and b) Eurasian needleleaf forests.

L 439: "It can be explained" – It is not clear to what "it" refers. I assume "it" refers to "L-VOD" from the previous sentence but then the sentence does not make sense. Please revise.
"It" refers to the whole previous sentence, *"We found that L-VOD is less impacted by fire than other VVs in this biome."* The pronoun "This" is better adapted. The correction was made.

L 502-503: I doubt this statement as it is written. You come to this statement by averaging over many fire events from several continents but for most individual fires, it will be likely very hard to see an effect in L-VOD because most fires are much smaller than the spatial resolution of L-VOD. You need to revise the sentence in a way that it is specific to the results of your study.
The reviewer is right, the sentence was modified by: *"Thus, L-VOD seems the best adapted vegetation variable for the monitoring of dense vegetation recovery after wide fires."*

Fig. 3: Some of the colours are almost impossible to see, especially the pale red and pale green of fires and L-VOD, respectively. What means "Fire nb" in the axis label?

In Fig. 3 and Fig. 5, we changed the pale red of fires by an orange colour; and the pale green of L-VOD by an intense green. Other colours were also modified for an enhanced visibility. An example is provided in Fig. R5 below. Please note that depending on your screen settings, colours may appear in a distorted way.
"Fire nb" means "number of fires". We replaced it with "fires".

[Figure]

Figure R5 - Time series of the number of fires, and anomalies time series of EVI, X-, C-, L-VOD, P, SM, TWS, and T, shifted on the fire date, for the needleleaf forest biome, with new colours.

Fig. 4: The time series plots are too small and the labels are not sharp.
We increased the size of the time series plots according to your suggestion (Fig. R6).

[Figure]

Figure R6 - Same as Fig. R2, modified for a better visibility.

**References**

Andela, N., Liu, Y. Y., van Dijk, A. I. J. M., de Jeu, R. A. M., and McVicar, T. R.: Global changes in dryland vegetation dynamics (1988–2008) assessed by satellite remote sensing: comparing a new passive microwave vegetation density record with reflective greenness data, Biogeosciences, 10, 6657–6676, https://doi.org/10.5194/bg-10-6657-2013, 2013.

Andela, N., Morton, D. C., Giglio, L., Paugam, R., Chen, Y., Hantson, S., Van Der Werf, G. R., and Randerson, J. T.: The Global Fire Atlas of individual fire size, duration, speed and direction, Earth Syst. Sci. Data, 11, 529–552, 2019.

Chuvieco, E., Aguado, I., Jurdao, S., Pettinari, M. L., Yebra, M., Salas, J., Hantson, S., de la Riva, J., Ibarra, P., Rodrigues, M., Echeverría, M., Azqueta, D., Román, M. V., Bastarrika, A., Martínez, S., Recondo, C., Zapico, E., and Vega, J. M.: Integrating geospatial information into fire risk assessment, Int. J. Wildland Fire, 23, 606, https://doi.org/10.1071/WF12052, 2014.

Fan, L., Wigneron, J.-P., Xiao, Q., Al-Yaari, A., Wen, J., Martin-StPaul, N., Dupuy, J.-L., Pimont, F., Al Bitar, A., Fernandez-Moran, R., and Kerr, Y. H.: Evaluation of microwave remote sensing for monitoring live fuel moisture content in the Mediterranean region, Remote Sens. Environ., 205, 210–223, https://doi.org/10.1016/j.rse.2017.11.020, 2018.

Forkel, M., Dorigo, W., Lasslop, G., Teubner, I., Chuvieco, E., and Thonicke, K.: A data-driven approach to identify controls on global fire activity from satellite and climate observations (SOFIA V1), Geosci. Model Dev., 10, 4443–4476, https://doi.org/10.5194/gmd-10-4443-2017, 2017.

Giglio, L., Randerson, J. T., and van der Werf, G. R.: Analysis of daily, monthly, and annual burned area using the fourth-generation global fire emissions database (GFED4), J. Geophys. Res. Biogeosciences, 118, 317–328, https://doi.org/10.1002/jgrg.20042, 2013.

Konings, A. G., Rao, K., and Steeleâ    Dunne, S. C.: Macro to micro: microwave remote sensing of plant water content for physiology and ecology, New Phytol., 223, 1166–1172, https://doi.org/10.1111/nph.15808, 2019.

Kuhn-Régnier, A., Voulgarakis, A., Nowack, P., Forkel, M., Prentice, I. C., and Harrison, S. P.: The importance of antecedent vegetation and drought conditions as global drivers of burnt area, Biogeosciences, 18, 3861–3879, https://doi.org/10.5194/bg-18-3861-2021, 2021.

Rogers, B. M., Soja, A. J., Goulden, M. L., and Randerson, J. T.: Influence of tree species on continental differences in boreal fires and climate feedbacks, Nat. Geosci., 8, 228–234, https://doi.org/10.1038/ngeo2352, 2015.

**References**

Broxton, P. D., Zeng, X., Sulla-Menashe, D., and Troch, P. A.: A global land cover climatology using MODIS data, J. Appl. Meteorol. Clim., 53, 1593–1605, https://doi.org/10.1175/JAMC-D-13-0270.1, 2014.

Giglio, L., Schroeder, W., & Justice, C. O. (2016). The collection 6 MODIS active fire detection algorithm and fire products. Remote Sensing of Environment, 178, 31-41.

Panciera, R., Walker, J. P., Kalma, J., & Kim, E. (2011). A proposed extension to the soil moisture and ocean salinity level 2 algorithm for mixed forest and moderate vegetation pixels. Remote Sensing of Environment, 115(12), 3343-3354.

Saleh, K., Wigneron, J. P., De Rosnay, P., Calvet, J. C., Escorihuela, M. J., Kerr, Y., & Waldteufel, P. (2006). Impact of rain interception by vegetation and mulch on the L-band emission of natural grass. Remote Sensing of Environment, 101(1), 127-139.

Togliatti, K., Hartman, T., Walker, V. A., Arkebauer, T. J., Suyker, A. E., VanLoocke, A., & Hornbuckle, B. K. (2019). Satellite L–band vegetation optical depth is directly proportional to crop water in the US Corn Belt. Remote Sensing of Environment, 233, 111378.

Togliatti, K., Lewis-Beck, C., Walker, V. A., Hartman, T., VanLoocke, A., & Hornbuckle, B. K. (2021, July). The B-Parameter Relating L-VOD to Satellite-Scale Crop Plant Water May Not Be Constant Over a Growing Season. In 2021 IEEE International Geoscience and Remote Sensing Symposium IGARSS (pp. 6857-6860). IEEE.

Wigneron, J. P., Calvet, J. C., De Rosnay, P., Kerr, Y., Waldteufel, P., Saleh, K., Escorihuela, M. J., & Kruszewski, A. (2004). Soil moisture retrievals from biangular L-band passive microwave observations. IEEE Geoscience and Remote Sensing Letters, 1(4), 277-281.

Wigneron, J. P., Schwank, M., Baeza, E. L., Kerr, Y., Novello, N., Millan, C., ... & Mecklenburg, S. (2012). First evaluation of the simultaneous SMOS and ELBARA-II observations in the Mediterranean region. Remote Sensing of Environment, 124, 26-37.

---

## Author Response (AR1)

Dear Associate Editor,
Thanks to the very useful comments of the reviewers, we improved the clarity of the manuscript and in particular the Introduction section and the justification of the selected biomes. The modifications can be found in red font in the track-changes file.
Thank you for your additional comments. Please find below in blue font our answers.
Moreover, we would like to inform the first reviewer (Dr. Chaparro) that after an investigation on several GDAL resampling methods, we decided to use a weighted average filter, for all datasets. Indeed, this resampling method is the most precise one to keep all contributions, and the results are quasi-identical to our previous method. We would like to thank Dr. Chaparro for raising this issue.
Best regards,
Emma Bousquet et al.

Dear authors,

Thank you for addressing the reviewer comments. As you can see, they each had a number of major concerns about the paper fully before publication can be considered. Please proceed to uploading the revised manuscript for re-review.

In addition, I have a few additional comments:

1) Like Reviewer 2, I had some trouble really understanding the goal of the study based on the somewhat meandering-introductions. From the list of planned individual changes it is difficult to assess how this will change, but please do try to ensure the objective of the study is clearly described and motivated.

We agree that the introduction was too general and did not emphasize the purpose of the study. We tried to simplify it and added the suggestions of both reviewers in order to focus on the objectives of the study.

2) In addition, it seems like it would still be possible to study post-fire recovery in areas that burn seasonally, just only by consider the recovery process on sub-annual timescales. Given the large number of areas with regular fires that are now not considered in what is profiled as a global study, why not add such an analysis?

In this study, we wanted to investigate the vegetation recovery in the long term, because fires can damage vegetation for many years in terms of AGB, as described in the introduction (Barlow et al., 2003; Silva et al., 2018; de Faria et al., 2021).
Moreover, a sub-annual study would be difficult to lead at the monthly timescale. We use here monthly averages of data in order to smooth rapid variations of VWC in VOD, as you are well aware (Konings et al., 2021).
Nevertheless, a sub-annual recovery study could be performed over seasonally burned areas in the future, by using daily or weekly data.

3) Given the dense vegetation in Santarem, is the soil moisture there really reliable?

Thank you for this very relevant comment. Soil Moisture measurements under dense forests are uncertain because of the lack of representative reference data for validation, even though Colliander et al. (2020) recently demonstrated that spaceborne L-band radiometry is indeed sensitive to soil moisture under forest canopies. We added TWS and precipitation data in order to support SMOS SM observations. Over Santarem area, no significant SM decrease is visible during the fire (Fig. 3c), while TWS and precipitation show a strong deficit. This surprising observation could be explained by the high uncertainties of SM under dense forest. We stated at line 312 : "Surprisingly, SM stayed stable during the fire." but we did not discuss this observation in the discussion. A sentence was added in that sense : "Contrary to TWS and precipitation, SM

stayed stable during the fire, maybe because of the reduced accuracy of SM measurements under very dense forest."

4) Can you add a sense of the spread across pixels to Figure 5? This should help contextualize the differences between sensors.

In Fig. 5, we show a mean anomaly time series over each biome (669 points for grasslands and croplands, 591 points for savannas and shrublands, 387 points for needleleaf forests, 79 points for sparse broadleaf forests, and 66 points for dense broadleaf forests). We chose to plot only the mean value without the dispersion because the resulting figure is illegible. An example is provided in Fig. R1 below for the dense broadleaf forest biome. The dashed lines represent the standard deviation. This information is interesting but the resulting graph is very heavy. Nevertheless, in order to provide an insight of the dispersion across pixels, we plotted the standard error of the mean of the measurements on Fig. 6 and Fig. 7.

[Figure]

Figure R1 – Time series of the number of fires, and anomaly time series of EVI, X-, C-, L-VOD, P, SM, TWS, and T, shifted on the fire date, for the dense broadleaf forest biome. Continuous line : mean value. Dashed lines : mean value +/- std.

5) Lines 461-462. Blaming the behavior of dense broadleaf forests in Figure 7 on a lack of fire-adaptation without further evidence is a big jump in logic. Aren't these anomalies more likely to be larger just because the mean VOD in these ecosystems before fire is larger?

The Associate Editor is right that different biomes are compared in Fig. 7, and that the mean value of VOD (and of other vegetation variables) is higher in dense forests. We found mean L-VOD values of 0.79 in dense broadleaf forests, 0.64 in sparse broadleaf forests, 0.61 in needleleaf forests, 0.36 in savannas and shrublands, and 0.21 in grasslands and croplands, in average over the considered points.
We tempered the corresponding sentence as follows : "We also found that the dense broadleaf forest biome was the most impacted by fire (Fig. 7), because the absolute values of vegetation variables before fires are higher in this biome, and because it is not a fire-adapted ecosystem (Cochrane, 2003).".

Best regards,
Alexandra Konings

**References**

Barlow, J., Peres, C. A., Lagan, B. O., and Haugaasen, T.: Large tree mortality and the decline of forest biomass following Amazonian wildfires, Ecol. Lett., 6, 6–8, https://doi.org/10.1046/j.1461-0248.2003.00394.x, 2003.

Cochrane, M. A.: Fire science for rainforests, Nature, 421, 913–919, https://doi.org/10.1038/nature01437, 2003.

Colliander, A., Cosh, M. H., Kelly, V. R., Kraatz, S., Bourgeau‑Chavez, L., Siqueira, P., et al (2020). SMAP detects soil moisture under temperate forest canopies. Geophysical Research Letters, 47(19), e2020GL089697.

de Faria, B. L., Marano, G., Piponiot, C., Silva, C. A., Dantas, V. D. L., Rattis, L., Rech, A.R., and Collalti, A.: Model-based estimation of Amazonian forests recovery time after drought and fire events, Forests, 12, 8, https://doi.org/10.3390/f12010008, 2021.

Greco, J. M.: Optical Radiation: Ultraviolet, Visible Light, Infrared, and Lasers. Hamilton & Hardy's Industrial Toxicology, 1045-1054, 2015.

Konings, A. G., Holtzman, N., Rao, K., Xu, L., and Saatchi, S. S.: Interannual Variations of Vegetation Optical Depth Are Due to Both Water Stress and Biomass Changes, Geophys. Res. Lett., 48, e2021GL095267, https://doi.org/10.1029/2021GL095267, 2021.

Silva, C. V., Aragão, L. E., Barlow, J., Espirito-Santo, F., Young, P. J., Anderson, L. O., Berenguer, E., Brasil, I., Brown, I. F., Castro, B., Farias, R., Ferreira, J., França, F., Graça, P. M. L. A., Kirsten, L., Lopes, A. P., Salimon, C., Scaranello, M. A., Seixas, M., Souza, F. C., and Xaud, H. A. M.: Drought-induced Amazonian wildfires instigate a decadal-scale disruption of forest carbon dynamics, Philos. T. Roy. Soc. B., 373, https://doi.org/10.1098/rstb.2018.0043, 2018.

---

## Referee Report (RR1)

**Review of the manuscript "SMOS L-VOD shows that post-fire recovery of dense vegetation is slower than what is depicted with X- and C-VOD and optical indices" (second review)**

I thank the authors for addressing part of my questions in the first revision round. I still have some major concerns with some of their answers, which should be addressed to improve the paper. The main issue is that the study is probably biased towards more humid regions (with drier ones such as the Sahel, the Mediterranean and Australia being under- or not represented). This is important because the consistency of the main conclusion (that recovery is slower as seen by L-band) is likely to be dependent on the climate and vegetation types (i.e., wetter regions have denser vegetation and require applying L-band VOD instead of C/X-VOD or VIS-NIR indices because L-VOD has greater penetration capacity). I detail my comments hereafter:

Major comments

1.  As the authors say, the number of fires is still low in the Sahel (due to the thresholds chosen). I also think that it is still low in the Mediterranean regions. It is important to include these regions in order to check whether your main conclusion holds in all cases or only in dense vegetation and/or mesic/humid sites. I think that this should be addressed in the manuscript. Can the authors provide a sensitivity analysis of the results according to these thresholds? In other words, how the changes in these thresholds impact the results? This could be addressed globally, or at least for the Sahel and Mediterranean regions. Should the authors adapt these thresholds in these regions? This should enable the inclusion of Sahel and Mediterranean regions. As the fire regimes are different around the globe, in some regions it may make not sense to keep the same threshold than in other regions.

2.  The results that the authors present comparing with and without Australia are interesting and show potential to improve the manuscript. I understand the exclusion of Australia, which is justified by (i) impossibility of a postfire study after 2019-2020 fires in forests, and (ii) the inclusion of the Outback fires unbalances the distribution. In that regard, point (i) has not potential solution and I agree that we need to wait until enough data is available. Regarding point (ii), I suggest two possible solutions:

    a.  The authors could choose a random subset of Australian shrub fires (with a balanced sample in comparison with the rest of fire events in the same land cover worldwide). Then include this subset in the global analysis.
    b.  The authors may leave the global analysis without Australia, but at least they should include a specific analysis of Australia in the supplementary and suggest possible interpretations. Not only because of completeness, but also because the results shown in the revision are interesting! I suggest some ideas hereafter:
        i.  The anomaly of temperature before fires when including Australia is lower. A possible interpretation of this is that, in dry regions and seasons (e.g., Australian summers), the occurrence of a large number of fires is not strongly linked to a positive temperature anomaly. Dry and warm conditions are normal for the season in that region, which means that standard summer conditions in the region are prone to wildfire ignitions most of the years. We found a similar result for summers in the Iberian Peninsula in Chaparro et al., 2016, where positive

temperature anomalies were helpful to predict fire ignitions mainly out of the summer season (i.e., we do not need anomalously warm summers in the Iberian Peninsula to be under high risk of fire ignition; the same should be expected in Australia, so I think that your results are consistent).

ii. The increase of the C- and X-band anomaly before fires could be caused by an accumulation of fuel (more biomass in shrublands). This is consistent with the fact that C- and X-bands could be more sensitive to biomass in open and small shrublands than L-band (due to lower penetration and lesser soil contamination of the VOD signal). But this should be further explored, maybe also using absolute values (not only anomalies) of P, T and EVI in the region, to understand if this interpretation is consistent.

iii. The authors find slower recovery in Australia. Is this due to a drier and warmer climate (i.e., more difficulty for recovery) than in other regions analyzed in the study? Considering that the Sahel and the Mediterranean are under-represented, this may also indicate that the study is shifted towards more humid climates, which is a reasonable conclusion looking at the fires' distribution map. This reinforces my argument in comment 1 (i.e., drier regions such as the Sahel and the Mediterranean should be included, as well as Australia).

Minor comments

- The authors' answer to my comment in lines 311-312 is very interesting and very well justified. I suggest that you include this explanation briefly in the paper.

References

Chaparro, D., Piles, M., Vall-Llossera, M., & Camps, A. (2016). Surface moisture and temperature trends anticipate drought conditions linked to wildfire activity in the Iberian Peninsula. European Journal of Remote Sensing, 49(1), 955-971.

---

## Author Response (AR2)

Dear Associate Editor, dear Reviewers,
Thank you for your additional comments and suggestions. You were right to state that this study was not global. We reframed our objectives and included many of your suggestions to improve the manuscript.
Please find below in blue font our answers, and in red font the changes against the previous version.
Sincerely,
Emma Bousquet et al.

**Associate Editor decision: Reconsider after major revisions**
**by Alexandra Konings**

Dear authors,

Thank you for your detailed responses to the reviews. As you can see, both reviewers are still concerned that your manuscript claims to be a global scale fire study but leaves out major fire hotspots like the Sahel, Australia, and, to a lesser but still significant extent, the European Mediterranean region. I agree with them. I do not fully understand why you believe that using monthly averages means you can't analyze the recovery at monthly timescales, which would enable you to add these regions in a relatively straightforward manner. Nevertheless, I understand the authors do not want to undertake this in the present manuscript. However, it is still necessary for a global study of fire recovery to actually consider global fire hotspots.

The two reviewers provide two different solutions to this mismatch between the papers' claimed coverage and its actual coverage. I agree with Reviewer 1 that the thresholds you use to exclude seasonal fires seem somewhat arbitrary, and you should check you can't include more regions by changing them locally. This would be the best solution. If this is not possible, I suggest following Reviewer 2's recommendation and reframing objective, abstract, introduction, and title to be clear that this study applies only in more humid sites, or at least in only a limited number of locations across the globe and why. This would require extensive rewriting.

We fully agree that calling the paper as if it was a global-scale study was probably inadequate, so we have modified it accordingly to address the Reviewers' comments. Consequently, we decided to: i) include Australia in the ecosystem-scale study; ii) reduce the time range for the studied fire events; iii) exclude areas prone to seasonal fires; iv) increase the second threshold in order to include more fire events. These choices are further explained below, in our answers to the Reviewers.

Regardless of which of these solutions you choose, I also agree with Reviewer 2's concern that the current title is highly technical and will turn away a large number of potential interested Biogeosciences readers who are not experts in microwave radiometry, not to mention underselling the novelty of the analyses. I suggest reframing as suggested.

We also agree with you and Reviewer 2, and decided to change the title as follows : "*Monitoring post-fire recovery of various vegetation biomes using multi-wavelength satellite remote sensing*".

Lastly, per journal policy, please ensure your figures are colorblind-friendly.

For that matter, two colorblind-friendly palettes designed by Paul Tol (2021) were used.

Best regards,
Alexandra Konings

**References**
Tol, Paul. 2021. "Colour Schemes." Technical note SRON/EPS/TN/09-002 3.2. SRON. https://personal.sron.nl/~pault/data/colourschemes.pdf.

Review of the manuscript "SMOS L-VOD shows that post-fire recovery of dense vegetation is slower than what is depicted with X- and C-VOD and optical indices" (second review)

I thank the authors for addressing part of my questions in the first revision round. I still have some major concerns with some of their answers, which should be addressed to improve the paper. The main issue is that the study is probably biased towards more humid regions (with drier ones such as the Sahel, the Mediterranean and Australia being under- or not represented).

This is important because the consistency of the main conclusion (that recovery is slower as seen by L-band) is likely to be dependent on the climate and vegetation types (i.e., wetter regions have denser vegetation and require applying L-band VOD instead of C/X-VOD or VIS-NIR indices because L-VOD has greater penetration capacity). I detail my comments hereafter:

Major comments

1. As the authors say, the number of fires is still low in the Sahel (due to the thresholds chosen). I also think that it is still low in the Mediterranean regions. It is important to include these regions in order to check whether your main conclusion holds in all cases or only in dense vegetation and/or mesic/humid sites. I think that this should be addressed in the manuscript. Can the authors provide a sensitivity analysis of the results according to these thresholds? In other words, how the changes in these thresholds impact the results? This could be addressed globally, or at least for the Sahel and Mediterranean regions. Should the authors adapt these thresholds in these regions?

This should enable the inclusion of Sahel and Mediterranean regions. As the fire regimes are different around the globe, in some regions it may make not sense to keep the same threshold than in other regions.

We fully agree with the Reviewer that our study is not global. Our goal was not to provide global results but to extend the analysis to a larger scale, which is the ecosystem scale. We decided to monitor only the major (i.e., large) and unique fires during the time period, in order to both be observed with the SMOS satellite (broad spatial resolution), and to avoid other perturbations (i.e., secondary fires) which can bias the analysis. With respect to seasonal fires, we consider that the vegetation cannot fully recover before the following fire event, that's why we decided not to keep them in the dataset. Indeed, in the savannas and shrublands of the Sahel (Fig. R1), we can observe a regular yearly decrease in the vegetation variables corresponding to the seasonal fires. The signals are very noisy and no major conclusion can be drawn from the observation of the major fire. If we mixed these time series with time series over areas showing a unique and major fire event, it would strongly dampen our observations. A sensitivity study was conducted to set the threshold values. Fig. 8 of the manuscript was also plotted for other biomes and led to a choice of 5 for the first threshold. Indeed, a compromise must be reached between a threshold high enough to observe a significant impact of fire; and low enough to keep a significant number of fire events in the sub-dataset. The second threshold enables to exclude areas subject to other major fires during the time period, and was arbitrarily fixed to 2. A similar trade off must be made in order to eliminate seasonal fires while keeping enough fire events. We agree to increase this threshold to 2.5,

which is half the first threshold, and which enables us to add several points to the dataset, in particular in areas affected by seasonal fires.
Moreover, we believe that it would be arbitrary to set different threshold values according to the biome, and we prefered to fix the same values for all biomes.

The text was modified in order to clarify our choices, and to remove the term "global":

L. 13: *"we investigated pre and post fire vegetation anomalies  over different biomes"*

L. 80: *"To evaluate the long-term impact and recovery, the study focused on areas with unique fire events, thereby excluding areas with regular seasonal fires where the vegetation cannot fully recover before the following fire event. Fire-prone areas are then excluded from this study. We first observed three particular cases of large fires and then extended the analysis for different biomes."*

L. 187: *"The rationale was to capture significant and unique events occurring over an area large enough to be observed with the SMOS satellite without any ambiguity."*

L. 253, 299, and 396: *"Extension to the  ecosystem scale"*

L. 254: *"Fires were then studied at the  ecosystem scale to assess the general factors and impacts of fire according to the specific features of each biome."*

[Figure]

Fig. R1 - Anomaly time series over the Sahel region, without applying a second threshold on the secondary fires => the anomaly of vegetation variables is quite stable, with a light decrease each year during the fires.

2. The results that the authors present comparing with and without Australia are interesting and show potential to improve the manuscript. I understand the exclusion of Australia, which is justified by (i) impossibility of a postfire study after 2019-2020 fires in forests, and (ii) the inclusion of the Outback fires unbalances the distribution. In that regard, point (i) has not

potential solution and I agree that we need to wait until enough data is available. Regarding point (ii), I suggest two possible solutions:

a. The authors could choose a random subset of Australian shrub fires (with a balanced sample in comparison with the rest of fire events in the same land cover worldwide). Then include this subset in the global analysis.

b. The authors may leave the global analysis without Australia, but at least they should include a specific analysis of Australia in the supplementary and suggest possible interpretations. Not only because of completeness, but also because the results shown in the revision are interesting! I suggest some ideas hereafter:

i. The anomaly of temperature before fires when including Australia is lower. A possible interpretation of this is that, in dry regions and seasons (e.g., Australian summers), the occurrence of a large number of fires is not strongly linked to a positive temperature anomaly. Dry and warm conditions are normal for the season in that region, which means that standard summer conditions in the region are prone to wildfire ignitions most of the years. We found a similar result for summers in the Iberian Peninsula in Chaparro et al., 2016, where positive temperature anomalies were helpful to predict fire ignitions mainly out of the summer season (i.e., we do not need anomalously warm summers in the Iberian Peninsula to be under high risk of fire ignition; the same should be expected in Australia, so I think that your results are consistent).

ii. The increase of the C- and X-band anomaly before fires could be caused by an accumulation of fuel (more biomass in shrublands). This is consistent with the fact that C- and X-bands could be more sensitive to biomass in open and small shrublands than L-band (due to lower penetration and lesser soil contamination of the VOD signal). But this should be further explored, maybe also using absolute values (not only anomalies) of P, T and EVI in the region, to understand if this interpretation is consistent.

iii. The authors find slower recovery in Australia. Is this due to a drier and warmer climate (i.e., more difficulty for recovery) than in other regions analyzed in the study? Considering that the Sahel and the Mediterranean are under-represented, this may also indicate that the study is shifted towards more humid climates, which is a reasonable conclusion looking at the fires' distribution map. This reinforces my argument in comment 1 (i.e., drier regions such as the Sahel and the Mediterranean should be included, as well as Australia).

Thank you for these comprehensive comments and suggestions.

Regarding option a, we believe that selecting a random subset of Australian shrub fires would be arbitrary compared to other regions. Option b is interesting, but we propose another way to include Australia to our study. Indeed, we stated that the main fires occurred in Australia at the beginning and at the end of the time period 2012-2020, which prevented a pre- and post-fire study. To circumvent this problem, we propose to reduce the time range of the studied fire events, by removing 14 months at the beginning and at the end of the period. The chosen time range for fire events is then Sept. 2013 – Oct. 2019, while keeping the longer time range (July 2012 – Dec. 2020) for the studied variables. This modification enables the inclusion of many cases occurring in Australia in the study, while avoiding an unbalanced spatial distribution of fires. The location and corresponding time series are shown in Fig. R2 and R3, for each biome.

NB : in Fig. R3, which corresponds to the updated Fig. 5 of the manuscript, we added the number of points in order to clarify that this number is not constant, as fires are artificially shifted to collocate in time all fire events.

Nevertheless, we added the interesting explanations that you provided to the manuscript :

Line 397: *"Grasslands, croplands, shrublands and savannas do not show signs of pre-fire drought (Fig. 5a, 5b, 6). Indeed, in these dry ecosystems, the standard summer conditions are often prone to wildfire ignitions (Chaparro et al., 2016)."*

Line 398: *"A substantial increase in vegetation variables, C- and X-VOD in particular, occurs 1 to 2 years before fire, which implies an increase in vegetation density, e.g. available fuel. This is consistent with the fact that C- and X-bands are more sensitive to dry low shrubland vegetation (Jackson et al., 1982; de Jeu et al., 2008)."*

L 82: *"Fire-prone areas are then excluded from this study."*

[Figure]

Fig. R2 - Location of the selected fires and histograms of the fire dates for each biome, with 1) fires from Sept. 2013 to Oct. 2019; 2) the inclusion of Australia; and 3) a first lower threshold of 5 fires and a second higher threshold of 2.5.

[Figure]

[Figure]

Fig R3 - Time series of the number of fires, and anomaly time series of EVI, X-, C-, L-VOD, P, SM, TWS, and T, shifted on the fire date, for the (a) grasslands and croplands biome; (b) savannas and shrublands biome; (c) needleleaf forest biome; (d) sparse broadleaf forest biome; and (e) dense broadleaf forest biome. Missing values appear when the number of available points is lower than half the maximum number of points of the biome (empty circles in the lower panel). This is mostly due to snow filtering. Data are kept otherwise (black filled dots).

Minor comments
• The authors' answer to my comment in lines 311-312 is very interesting and very well justified. I suggest that you include this explanation briefly in the paper.

Thank you for your suggestion. We modified the sentence at line 290 : "*Strong positive temperature anomalies (+3°C), negative precipitation anomalies (-160 mm) and TWS*

*anomalies (-60) are visible  during the fire, and reach their maximum at the end of the fire period.*"

The explanation is then provided in the Discussion, at line 385 : "*These extreme drought conditions worsened during and at the end of the fire, and may explain its strength. Several factors can explain this observation. First, MODIS may not detect all fires in Jan. 2016 in this area, because of i) the cloud coverage (Roy et al., 2008) and ii) the dense vegetation cover hiding understory fires (Withey et al., 2018). This would be in line with the 2016 Hansen et al. tree cover loss detection (Fig. 2). Secondly, drought may sometimes keep increasing after fire extinguishment, because the removal of the vegetation cover and the deterioration of the soil contributes to maintaining a hot and dry climate (Auld and Bradstock, 1996; Veraverbeke et al., 2010). This phenomenon is also visible in the savanna and in the sparse broadleaf biome (Fig. 5b and 5d).*"

References
Chaparro, D., Piles, M., Vall-Llossera, M., & Camps, A. (2016). Surface moisture and temperature trends anticipate drought conditions linked to wildfire activity in the Iberian Peninsula. European Journal of Remote Sensing, 49(1), 955-971

Dear authors,

Thank you for carefully addressing the comments by the two reviewers and the associate editor. I feel that most of you revisions and responses are appropriate.

However, I still think that the objective of the study should be stated more clearly in the abstract and the introduction. Thereby you also should make clear from the beginning that you focus on regions with a low fire frequency/without seasonal fires (but specify how you define "low" fire frequency). For example, you could write "Our objective is to quantify and compare the post-fire vegetation recovery based on visible-infrared EVI and microwave X-, C- and L-VOD for selected fire events and for different biomes."

The reviewer is right, our study is not global. We clarified this in the manuscript :

L. 13: *"we investigated pre and post fire vegetation anomalies*  *over different biomes"*

L. 80: *"To evaluate the long-term impact and recovery, the study focused on areas with unique fire events, thereby excluding areas with regular seasonal fires where the vegetation cannot fully recover before the following fire event. Fire-prone areas are then excluded from this study. We first observed three particular cases of large fires and then extended the analysis for different biomes."*

L. 186, 300 and 484: *"* *ecosystem scale"*

L. 187: "*The rationale was to capture significant and unique events occurring over an area large enough to be observed with the SMOS satellite without any ambiguity."*

L. 254, 299, and 396: *"Extension to the*  *ecosystem scale"*

L. 255: *"Fires were then studied at the*  *ecosystem scale to assess the general factors and impacts of fire according to the specific features of each biome."*

Personally, I'm not really happy with the title of your study. The title is too focussed on a single element of your results and is with using four different abbreviations (i.e. SMOS, L-VOD, X-VOD, C-VOD) very technical and not very attractive for a non-specialist audience. Your results actually nicely demonstrate the different recovery times of different vegetation variables in different ecosystems and hence are much richer than what the title is suggesting. In addition, for the specialist VOD-related community the title is rather obvious as L-VOD is known to be more sensitive to woody components than shorter wavelengths and it is obvious that woody components recover slower than grass or shrubs. Hence as a VOD-specialist, I find the title not very novel and as a general fire/vegetation scientist, I feel discouraged from the technical jargon. I suggest to better reflecting the richness and diversity of your results in the title, which could make the paper more attractive for a wider audience.

We agree with the Reviewer and decided to change the title as follows : "*Monitoring post-fire recovery of various vegetation biomes using multi-wavelength satellite remote sensing*".

L262-264: "For that, a minimum threshold of 5 was applied on the maximum number of fires; and a maximum threshold of 2 was applied outside the main fire event period (i.e. outside the period -6/+6 months around the fire event)." - This sentence is very difficult to understand. Does the "maximum number of fires" refer to the number of fires in a month? But why "maximum"? Does this mean that you selected fire events if more than five fires occurred in a month but only if less than two fires occurred in the 6 months before or after? Please try to revise this sentence.

Yes, absolutely. We rephrased this sentence in order to clarify its meaning:
*"Two conditions were empirically defined as mandatory to select a fire event over a given pixel: i) a minimum number of fires of 5 at the height of the fire; ii) a maximum number of fires of 2.5 outside the period [-6/+6] months around the main fire event, to ensure that the vegetation recovery is linked with the main fire event and is not affected by another significant one."*

Figure 5: The figure is still very small (almost not readable at 100%) and I'm wondering if it can be really improved during the final edits. Maybe consider to rearrange the figure or split it over two pages in order to improve it.

Indeed, we wish to split Fig. 5 over two pages in order to be more readable. Colours were also changed with a colorblind-friendly palette designed by Paul Tol (2021).

Best regards,
Matthias Forkel

**References**

Tol, Paul. 2021. "Colour Schemes." Technical note SRON/EPS/TN/09-002 3.2. SRON. https://personal.sron.nl/~pault/data/colourschemes.pdf.

---

## Author Response (AR3)

Dear Associate Editor and Reviewer,

We are delighted to learn that our manuscript is now acceptable for publication.

We took into account the minor comments of Dr. Chaparro regarding the text and modified the manuscript accordingly. As regards TWS data, they should be in centimetres indeed. We modified the figures and the text accordingly.

We thank you again for your relevant corrections and suggestions during the whole process.

Sincerely,

Emma Bousquet et al.